# Vapor-induced phase-separation-enabled versatile direct ink writing

Marc Sole-Gras [1], Bing Ren[1], Benjamin J. Ryder [2], Jinqun Ge[3], Jinge Huang[4], Wenxuan Chai[1], Jun Yin [5], Gerhard E. Fuchs[2], Guoan Wang[3], Xiuping Jiang[4] & Yong Huang [1,2] ✉

Versatile printing of polymers, metals, and composites always calls for simple, economic approaches. Here we present an approach to three-dimensional (3D) printing of polymeric, metallic, and composite materials at room conditions, based on the polymeric vapor-induced phase separation (VIPS) process. During VIPS 3D printing (VIPS-3DP), a dissolved polymer-based ink is deposited in an environment where nebulized non-solvent is present, inducing the low-volatility solvent to be extracted from the filament in a controllable manner due to its higher chemical affinity with the non-solvent used. The polymeric phase is hardened in situ as a result of the induced phase separation process. The low volatility of the solvent enables its reclamation after the printing process, significantly reducing its environmental footprint. We first demonstrate the use of VIPS-3DP for polymer printing, showcasing its potential in printing intricate structures. We further extend VIPS-3DP to the deposition of polymer-based metallic inks or composite powder-laden polymeric inks, which become metallic parts or composites after a thermal cycle is applied. Furthermore, spatially tunable porous structures and functionally graded parts are printed by using the printing path to set the inter-filament porosity as well as an inorganic space-holder as an intra-filament porogen.

Wide implementation and adoption of material extrusion for additive manufacturing (AM)/three-dimensional (3D) printing applications are still limited by two main factors: the accessibility of starting build materials, and the availability of feasible phase change mechanisms for material dispensing and subsequent solidification upon deposition. As such, extrusion-based printing is mostly used for thermoplastics in terms of fused filament fabrication. As a variant of extrusion-based AM, direct ink writing (DIW) expands the selection of build materials for extrusion-based AM. During DIW, the ink is usually dispensed through a nozzle in a controlled manner, following a predefined path to conform structures in a layer-by-layer fashion[1]. A key DIW step is to introduce in situ rapid solidification mechanisms to the build material

being printed, which include photopolymerization[2], fluid-to-gel transition by pH change[3], temperature change[4], polyelectrolyte exchange reaction[5], and thermodynamic instability-induced phase separation[6–8], to name a few. Of them, the phase-separation mechanism[6–8] relies on the demixing of an initially homogeneous polymeric solution into polymer-rich and polymer-lean phases upon induction of an instability by the presence of non-solvent or by a change in temperature, providing an alternative solidification venue for polymer printing. For 3D printing applications, this can be accomplished by immersion phase separation[8] and solvent evaporation-based phase separation[6,7]. While such a polymer printing approach is convenient and only requires a polymer to be dissolvable, it still entails some challenges from a

[1]Department of Mechanical and Aerospace Engineering, University of Florida, Gainesville, FL, USA. [2]Department of Materials Science and Engineering, University of Florida, Gainesville, FL, USA. [3]Department of Electrical Engineering, University of South Carolina, Columbia, SC, USA. [4]Department of Food, Nutrition, and Packaging Sciences, Clemson University, Clemson, SC, USA. [5]School of Mechanical Engineering, Zhejiang University, Hangzhou 310027, China. ✉e-mail: yongh@ufl.edu

perspective of printability, process, and material selection: the use of immersion bath printing may require additional post-processing to fully remove the supporting phase[9], or the solidification rate of deposited materials does not usually allow for proper adjacent filament fusion during printing as the phase-inversion kinetics are bound by the fixed amount of non-solvent in the printing environment or the evaporation rate of volatile solvent. By overcoming the aforementioned challenges, herein we demonstrate a DIW process at ambient conditions based on the use of polymer as a build material or powder binder and the vapor-induced phase-separation process (VIPS) as a polymer solidification mechanism. A versatile AM platform is developed accordingly and further extended to the printing of composite materials by using polymer-based composite suspension inks, and the polymer phase can be removed during a post-processing step. We envision that the VIPS-based 3D printing process expands the availability of build materials ranging from polymers to metals to composites as well as the feasibility of printable multi-scale porous structures with a functional gradient, which are not readily achievable by current manufacturing approaches.

## Results

### VIPS-enabled 3D printing process

Herein, we report a 3D printing technology that utilizes vapor-induced phase separation (VIPS) to 3D print polymer- and composite-based parts. VIPS is a type of non-solvent-induced phase separation (NIPS), used here as the in situ solidification mechanism, circumventing the aforementioned challenges associated with typical phase inversion-based printing processes. During printing, the thermodynamic instability that provokes a homogeneously dissolved polymer mixture to separate and demix into two different phases is based on the penetration of a non-solvent, in its vaporous form, into the polymer

and solvent solution and the difference of their degrees of chemical affinity, which is one of the main phase inversion methods for ternary polymer/solvent/non-solvent systems[10–12]. The resulting phases from the phase-separation and demixing process include polymer-lean and polymer-rich ones, where the formation of the latter is responsible for the solidification and hardening process of the extruded material using this printing technology. The proposed printing process, named VIPS-3DP (vapor-induced phase-separation 3D printing), can be easily expanded beyond the printing of polymeric inks, where colloidal inks of polymer solution-based metallic and ceramic suspensions are successfully 3D printed in air without the need for auxiliary support.

During VIPS-3DP, the presence of vapor or mist of non-solvent is used to induce polymeric phase separation and demixing that serves as an in situ solidification mechanism occurring within the newly deposited filaments, which are comprised of a mixture of polymer, non-toxic, low vapor pressure solvent, and/or other powders, porogens, and additives, allowing the 3D structure buildup (Fig. 1a). The presence of the vaporous non-solvent results in solvent/non-solvent exchange dynamics that serve as the driving force for solidification due to the increased presence of a polymer-rich phase upon demixing. The longer the amount of time exposed to the non-solvent environment, the higher the filament stiffness due to the larger quantity of solvent being extracted from the deposited volume as the result of the mass exchange between the solvent diffusing out of the deposited ink and vaporous non-solvent diffusing into the ink. Depending on the induced mass exchange kinetics between the solvent and non-solvent, the deposited filament may present a porous morphology (inset, left side). The outer region of the deposited filament is the first one to solidify by being exposed first to the nebulized non-solvent, and thus the solidification front travels inwards, the innermost region being the last one to demix and solidify (inset, right side). From a 3D printing

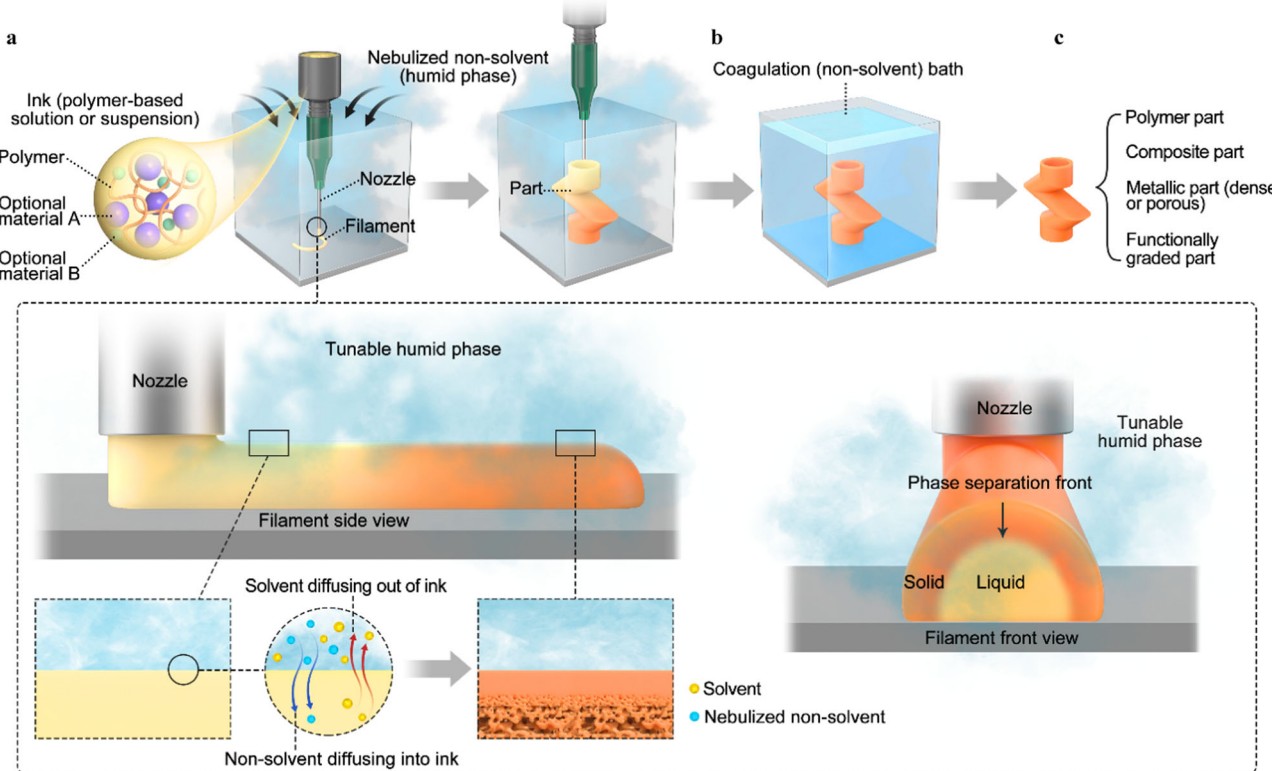

**Fig. 1 | Schematic of VIPS-3DP process. a** Printing under nebulized non-solvent from a part perspective. The inset shows a detailed view from a filament perspective: clear colors indicate material concentrations as deposited and no solidification present; darker orange colors indicate a higher solidification and stiffness degree due to solvent-vaporous non-solvent exchange dynamics, which generally results in a filament morphology with a dense outer layer and porous inner sections. **b** Complete solidification in coagulation bath and **c** versatile printed parts: polymer, composite, metallic, and functionally graded.

perspective, the ideal solidification process should guarantee enough stiffness to allow the structural buildup of new material while retaining enough dissolved polymer to ensure adequate interlayer fusion in adjacent filaments. The solidified outermost layer of a previous filament is partially dissolved by the solvent of a consecutively deposited filament, forming a good fusion bond. Moreover, the phase-inversion kinetics are bound by the chemical affinity of the compositions of the ternary system and the amount of vaporous non-solvent present that determines the rate at which the deposited filament solidifies. That results in more efficient printing enabled by faster solidification dynamics to occur under nebulized conditions, such as relative humidity (RH) ≈ 99% using water as a non-solvent, and with materials having a lesser chemical affinity to the solvent as seen from the Hansen's relative energy difference (RED) values between the polymer and solvent. Given that water is classically used as the non-solvent and its amount plays a key role in the phase-separation kinetics, the solidification rate can be tuned by changing the environmental RH: under room conditions with a relative humidity of approximately 40% or using a water nebulizing system that delivers water mist in the printing region to yield RH values in a controlled manner that can range approximately from a room condition to 99%.

To ensure complete solidification throughout the part, the printed parts are immersed in a non-solvent-based coagulation bath (Fig. 1b) and the porogen material, if present, is dissolved to obtain a porous part. The resulting solvent can be collected and further reclaimed accordingly for environmentally conscious manufacturing. It is also expected that solvents with a low vapor pressure should be utilized to avoid potential evaporation during printing and applicable hydrophobic or hydrophilic substrate surface should be used for better collection of the resultant solvent and non-solvent mixture upon printing without compromising printed structures. A wide variety of parts can be printed (Fig. 1c) based on the solidification mechanism of polymeric phase separation: single to multiple polymers-based parts and composite parts. VIPS-3DP composite parts can be classified into two different categories as defined by whether the final product includes the polymeric material or not. If the polymeric material is to be removed, it functions as a binder during the 3D printing of green parts and is later removed, usually via a thermal cycle that aims to pyrolyze or burn out the binder-acting polymer and inducing solid-state sintering between the metal or ceramic phase. Using such an approach, quality metal and ceramic parts are also able to be fabricated.

## Polymer printing

As the first demonstration, polymer parts are printed using acrylonitrile butadiene styrene (ABS) and polyacrylonitrile (PAN), two model polymers, using VIPS-3DP. The model solvent used herein is dimethyl sulfoxide (DMSO) because of its nontoxicity, nonvolatility due to its low vapor pressure and its easiness to be separated from the water used herein as the non-solvent, via a simple distillation process. From an environmental perspective, the use of low-volatility solvents enables their reclamation after the printing process, significantly reducing the environmental footprint as well as the fume exposure to users and equipment during printing when compared to other evaporation-based phase-separation processes. The increase of the non-solvent concentration in the printed volume is obtained by the delivery of water mist in the printing region. The polymers are selected based on the dissolution feasibility with the solvent of interest, which is easily assessed using Hansen's solubility theory and its RED values (Supplementary Tab. 1). Both polymers have good dissolution in DMSO but different chemical affinity degrees with the solvent (DMSO) as well as with the non-solvent (water), which heavily influence the mass transport kinetics of the solvent/non-solvent exchange, playing a key role in the porosity and morphology of final parts[13]. The stability boundaries of the system comprised of the polymer, solvent,

and non-solvent can be graphically expressed using a ternary phase diagram (Supplementary Table 2, Supplementary Fig. 1), known as the binodal curve, where it can be observed that PAN presents a larger stable, one-phase region when compared to that of ABS. This leads to slower demixing kinetics of PAN upon exposure to the non-solvent (water) as seen from the different polymer-solvent interaction parameters $g_{23}$. Typical evolution of the solidification front after filament printing can be seen in Supplementary Figs. 2 and 3.

Figure 2a presents the sequence of the printing of an ABS-based vase-like structure as printed in air and without nebulized water (RH = 40%). One of the key advantages of using VIPS as a solidification mechanism is that such solidification dynamics can be adjusted by controlling the delivery of non-solvent to the printing area, specifically, the RH level herein. Printing under room conditions results in overall slower phase-separation dynamics when compared to that under a nebulized printing zone, as the amount of water in the environment is significantly different (RH: 40% vs 99%). For a given polymeric ink, the printing behavior is highly related to the increased RH from ink spreading to well-defined filament to deformed filament as exemplified in Fig. 2b. At low RH levels, the ink is not solidified quickly enough, and thus, there is enough time for the ink to deform due to its own weight and the lack of sufficient surface tension and hardening degree. This effect is most apparent for the low viscosity ink compositions such as 20% (w/v) ABS since these samples have low shear moduli resulting in poor stability. On the other hand, higher levels of RH may induce premature solidification-induced defects such as wavy filament shapes due to the fast solidification dynamics in place. Results show that 20%–60% (w/v) ABS inks under 40% and 70% RH have a larger filament diameter than that of the nozzle (herein a gauge 15 nozzle with an inner diameter of 1.36 mm) as the insufficient solidification after extrusion results in poor shape fidelity, as seen in Fig. 2c. More detailed printability and print shape accuracy studies can be found in Supplementary Figs. 4–6. As observed, the good combination of RH and polymer concentration is 70% RH and 30%–40% (w/v) ABS during VIPS-3DP ABS printing. For the demonstration of high-resolution printing using VIPS-3DP, a miniature ABS lattice is printed as shown in Fig. 2d, and each segment is approximately 0.30 mm tall (three layers for 0.10 mm each) and 0.30 mm wide (two filaments).

For effective 3D printing, deposited filaments should be strong enough to self-sustain the forming structure and allow further buildup while having a low solidification state to ensure good interlayer fusion between consecutive depositions. During polymer printing, the polymer demixing speed influences the permissible printing speed, and it is required that successive layers bond successively to ensure the part integrity as exemplified in Fig. 2e (the red arrow denotes the filament height). The ABS phase separates in a faster manner, so a higher deposition rate is feasible when compared to PAN. For instance, ABS-based inks are deposited with nozzle speeds of 2.0–5.0 mm/s while PAN-based inks are deposited with speeds of 0.5–2.0 mm/s during printing patterns with a 10–15 mm width. The lower-bound speeds are under room conditions, and the higher-bound speeds are in a nebulized environment. The solidification dynamics also play an important role in determining the final microstructure and interlayer fusion of the deposited filaments as shown in detail in Supplementary Figs. 7–10. Generally, two distinct morphologies are herein observed: faster solidification rates yield in porous asymmetric morphologies, while non-porous features are found during slower solidification kinetics that results in a denser, shrunk part. An example of asymmetric porosity is shown in Fig. 2f, where larger pores are observed at the bottom of the filament (red dashed line). By adjusting the phase-separation rate via the selection of materials and/or relative humidity present, different porous morphologies can be obtained. The fast solidification behavior can be further extended to print overhang features without additional support structures or baths. As shown in Fig. 2g and Supplementary Movie 1, a continuous filament is deposited

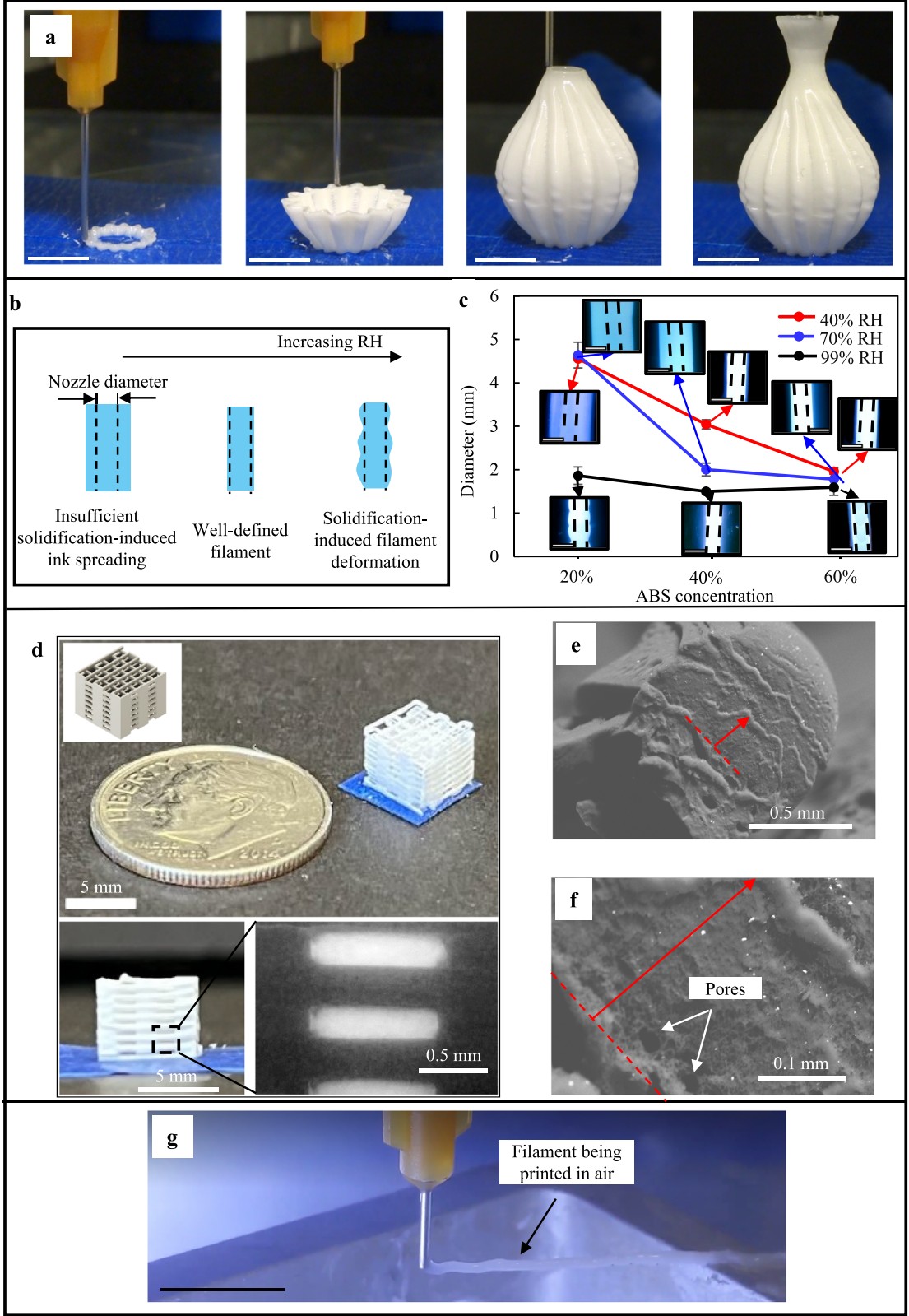

**Fig. 2 | ABS-based printing results. a** Sequence of a vase-like structure being printed, **b** schematics of different printed filament morphologies, **c** effects of ABS polymer ink concentrations (20%–60% w/v) on the printed filament diameters under 40%, 70%, and 99% RH and a 1.5 mm/s printing speed (Inset: images of the solidified filaments, and dashed lines in the inset: nozzle diameter. Scale bars: 2 mm), **d** a VIPS-3DP printed lattice part, **e**, **f** morphology details of polymeric cross sections, and **g** overhang filament printing process. (Scale bars in **a** and **g**: 10 mm, and the error bar is ± sigma if not specified).

in a nebulized environment, solidifying fast enough that the filament can maintain its horizontal shape as a cantilever beam.

## Metal printing

Furthermore, the polymeric phase found in the developed polymeric inks can be used as a binder agent to facilitate the printing of metallic powders when using polymer-metal suspensions as inks. Metallic parts can be produced by removing polymer from green parts using a post-printing thermal debinding and sintering stage similar to typical metal injection molding (MIM) or binder jetting processes. As shown in Fig. 3a, the printing process is similar to the printing of polymeric inks. Powders herein are suspended in the dissolved polymer with a volumetric ratio within the range of 50%–55%, close to the random close-packing (RCP) ratio of hard, equally sized particles of 64%[14]. Upon deposition, the printed parts undergo a thermal cycle under vacuum to remove the binder materials and sinter such parts. The debinding conditions are selected based on thermogravimetric analysis as detailed in Supplementary Fig. 11. The process of densifying the metallic particles is based on the atomic diffusion via solid-state sintering[15] by heating the brown part to a temperature of 70%–95% of the melting point of the metal or solidus point of the alloy (Supplementary Figs. 12 and 13, Supplementary Table 3). One of the key advantages of VIPS-3DP when compared to binder-based metal printing technologies is the convenient, low-cost printing setup at room temperature and the ability to efficiently tune the composition of the deposition by easily changing the input composition (e.g., integrating an active pre-printing ink mixer).

A wide variety of austenitic stainless-steel 316L parts have been printed for demonstration purposes, ranging from a Y-shape tube (Fig. 3b), tubular parts with a varying cross section (Fig. 3c), a lattice part with a varying diameter (Fig. 3d), a hip implant-like part (Fig. 3e), and cylindrical parts with hollow channels (Fig. 3f). The densification degree of these parts is found to be in the 92%–99% range when compared to the theoretical value, and this densification value is comparable with that of printed stainless-steel parts using metal injection molding (94%–96%)[16] and selective laser sintering (92%–99.6%)[17]. The printed and sintered 316L parts have an ultimate tensile strength (UTS) of approximately 475 MPa (Fig. 3g), which is comparable to those of the wrought and powder metallurgy routes such as MIM[18]. As such, the printed and sintered parts can undergo typical post-processing steps such as sandblasting, cutting, or tap threading (Supplementary Fig. 14). Figure 3 also shows the as-printed (Fig. 3h), as-sintered (Fig. 3i), and as-post-processed (sand-blasted) (Fig. 3j) turbine-like parts. The inner morphology of the parts shows how the polymer successfully binds powder particles at the green body stage (Fig. 3k), such polymer is removed by the thermal cycle to obtain a brown body as only metallic powders are observed (Fig. 3l), and a dense morphology is observed upon sintering (Fig. 3m). The sandblasting operation successfully removes the outer layer of the sintered part formed by debinding and sintering debris, presenting a smooth and shining surface texture without compromising the integrity of the parts that include thin blade-like features. The near-theoretical density of the steel allows the thin features to withstand the sandblasting process. Herein, ABS is used as a preferred binder due to its low decomposition temperature of 420–470 °C (Supplementary Fig. 11), which potentially widens the usability of low melting point materials such as zinc or aluminum. The part porosity is affected by the binder used as a result of the phase-separation behavior of polymers used. Usually, polymers with slower demixing kinetics may result in green parts with a low porosity value. As observed, metallic ABS-bound parts may result in porosity values of approximately 8%, but PAN-bound parts do not present any notable pores (Supplementary Fig. 13a, b). It should be noted that the use of PAN, a widely used carbon fiber precursor[19], may result in undesirable carbide formation due to its

carbon-rich composition at higher temperatures of 1000 °C[20] while ABS does not present such compositional behavior (Supplementary Fig. 13c, d).

Some functional stainless-steel parts are also printed. Figure 3n shows a K-band horn antenna with a gain of 12 dB at the best operational frequency at 22 GHz, and its technical specifications can be easily adjusted by printing different designs. Such manufacturing flexibility includes depositing overhang geometries with an angle of 45°. Additional information on the design, assembly, and testing process is available in Supplementary Figs. 15–17. The simulated and measured results of the return loss of the antenna are shown in Fig. 3o. The printed antenna has a −10 dB bandwidth of 4.4 GHz, and the lowest reflection coefficient ($S_{11}$) is measured as −36.58 dB at 24.5 GHz. The measured results present behavioral agreement with the simulated results with good performance from 22.3 GHz to 26.7 GHz, and the main sources of error are due to the step-like roughness in the inner surface and inevitable characterization errors by the setup utilized. Moreover, a tap threading process has been applied to a 3D printed and sintered zig-zag tube where a thread of 1/8 NPT (national pipe taper) has been added to one tube end (Fig. 3p). A push-to-connect fitting is assembled using the tap thread process, and no liquid leak is observed while blue-dyed water flows through (Supplementary Movie 2), proving that no significant open porosity is present.

## Composite printing

The use of VIPS-3DP is also investigated in terms of the feasibility to print various composites, ranging from polymer-metal to metal-ceramic composites, for which the polymer binder may or may not be removed depending on the application needs.

**Polymer-metal composite.** First, the printing of polymer-metal composites is investigated to fabricate copper-embedded polymeric structures for biocidal applications. Copper (Cu), either non-oxidized or oxidized as Cu particles[21], is a bioactive material due to its ability to rapidly inactivate fungal, bacterial, and viral pathogens by generating free ions in form of hydroxyls as a result of redox processes[22] that may induce damage to cellular membranes, proteins and DNA structures[23]. A scaled-down biocidal personal face respirator loaded with Cu particles is printed without any support structure using a polymer-metal ink with 60% (w/v) of ABS, 30% (w/v) of thermoplastic polyurethane (TPU), and 10% (wt%) Cu as shown in Fig. 4a, and its bacterial inhibition performance is shown in Fig. 4b, where a significant reduction in the growth rate is observed for *Staphylococcus aureus*, a model respiratory disease-related Gram-positive bacterium[24]. Cu-laden samples show a notable inhibition of the growth rate, being 8.2 $\log_{10}$ CFU/mL when compared to the material controls (9.0–9.2 $\log_{10}$ CFU/mL). The presence of polymeric phase-separation-induced porosity results in additional access to the surfaces of biocidal Cu particles for bacterial inactivation by contact. The sample reusability is studied upon ethanol cleaning, which results in identical growth inhibition features. Such reused samples are considered to contain copper oxide particles as noted by the change of color in the broth medium. The antibacterial rate, calculated as the % reduction to the control group in terms of CFU, is within the range of 92%–95% based on the small samples (~0.004 g) and the bacterial culture medium of 1 mL. By increasing the number of samples (such as two) to the testing medium, better performance of bacterial inhibition is observed with a 7.4 $\log_{10}$ CFU/mL value and an antibacterial rate of 98 %. Overall, the printed polymer-metal composite parts have significant antibacterial performance upon 12 h of culturing.

**Metal-ceramic composite.** Second, the production of metal-ceramic composites is demonstrated in printing stainless-steel 316L-hydroxyapatite (HAp) and nickel-tungsten carbide (WC) parts. By using VIPS-3DP, bioceramic HAp[25] can be easily included as part of the printing

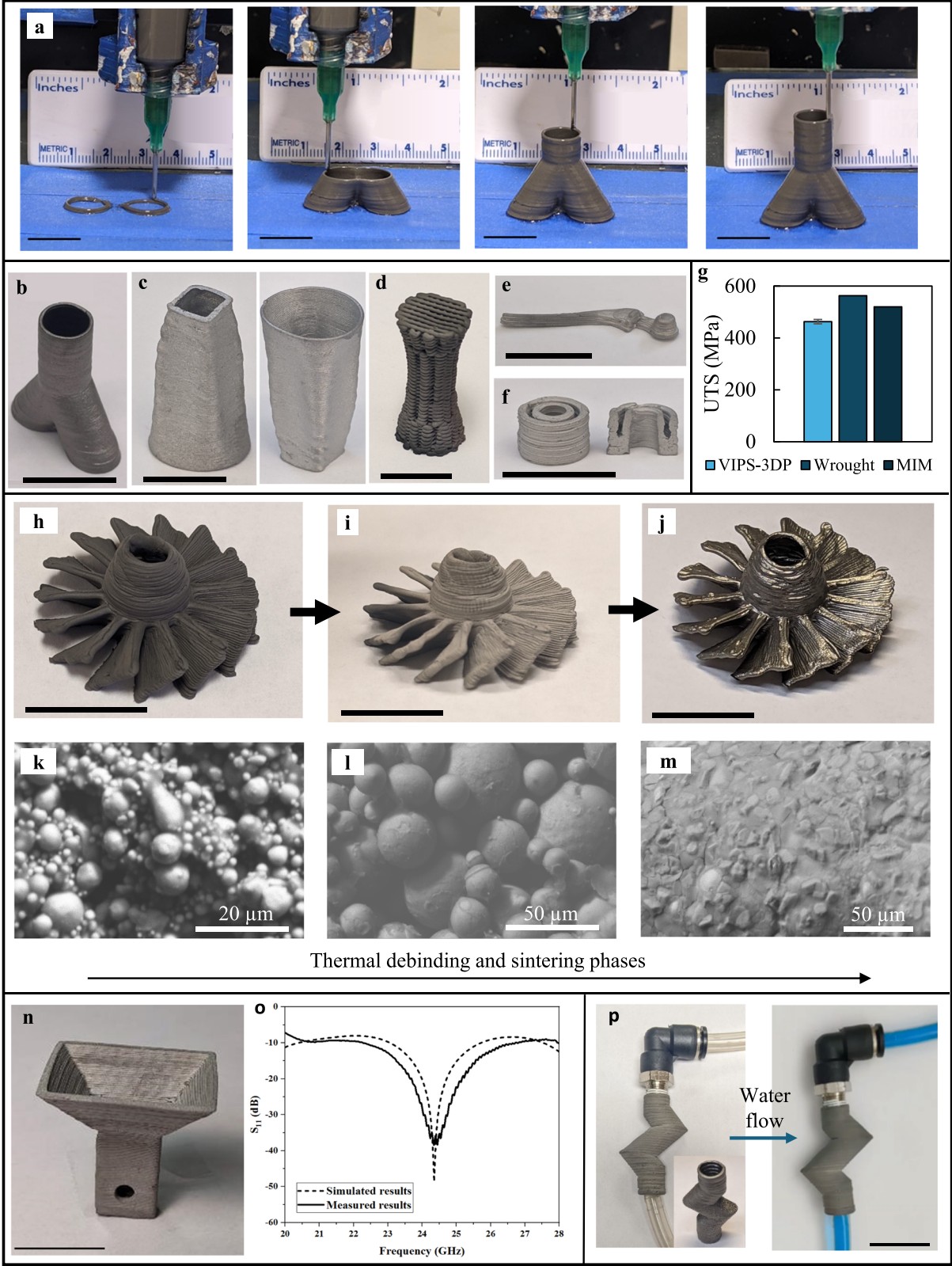

**Fig. 3 | Illustrative process of VIPS-3DP for stainless-steel 316L parts. a** Printing sequence of *Y*-shaped tubular part using VIPS-3DP. A variety of printed structures after sintering: **b** *Y*-shaped tube, **c** tube with a varying cross section, **d** lattice structure with a varying diameter, **e** scaled-down hip implant, and **f** cylinder with hollow channels. **g** UTS results. Turbine-like part: **h** as-printed, **i** as-sintered, and **j** as-post-processed (sandblasted) and **k**–**m** detailed morphologies of printed part in the green, brown, and sintered body states. **n** Printed and sintered horn antenna and **o** its reflection coefficient S$_{11}$ results. **p** Printed and sintered zig-zag tube with dyed water for leakage test. (Scale bars = 15 mm unless specified otherwise, and the error bar = ± sigma).

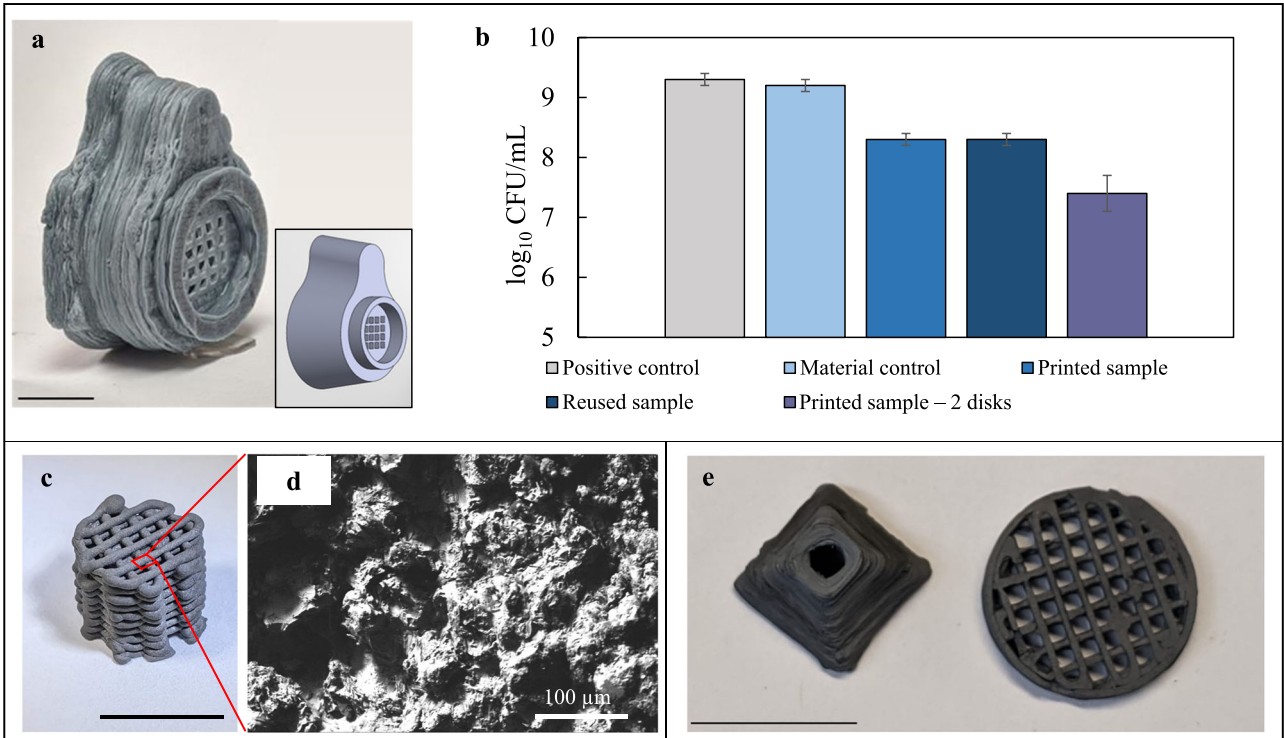

**Fig. 4 | VIPS-3DP composite printing results. a** Composite-based respirator (ABS-TPU polymer mixture-copper) and **b** *Staphylococcus aureus* inhibition results. **c** 316L-HAP composite scaffold and **d** densified cross-sectional details. **e** WC-Ni composite: pyramid and circular lattice parts (Scale bars = 10 mm unless specified otherwise, and the error bars = ± sigma).

process of 316L-HAp scaffolds for bone tissue engineering applications. The addition of 316L aims at avoiding poor HAp adhesion issues as reported during HAp coating processes[26]. 316L-HAp (5 wt%) composite scaffolds are shown in Fig. 4c, which combine the excellent mechanical properties of the steel and the great biocompatibility of HAp. HAp may undergo a transformation process that results in the presence of tricalcium phosphates (TCP) and tetracalcium phosphates (TTCP) during sintering[27]. The addition of HAp in small quantities does not affect the densification behavior of the steel phase as the cross-sectional area presents a dense morphology (Fig. 4d). The effects of volumetric densities of 316L and HAp on the mechanical properties are available elsewhere[28]. Energy-dispersive X-ray spectroscopy (EDS) results confirm the presence of calcium and phosphorus in small amounts (Supplementary Fig. 18). Moreover, VIPS-3DP is used to investigate the feasibility of the production of other alloys that aim at high-temperature applications. Refractory materials are considered hard-to-sinter due to their inherent temperature resistance. Fortunately, the easiness of mixing up different materials using VIPS-3DP opens up the possibilities for the development of self-infiltrating printed parts, whose porosity upon sintering due to lack of densification is notably attenuated by the use of a lower melting metal that undergoes a melting process[29]. This way, both solid- and liquid-state sintering occur in a simultaneous manner, and the liquified metal fills the voids due to the capillarity effect. Herein, WC-Ni composites (97(WC)-3(Ni) wt%) are printed and sintered as examples of self-infiltrating parts as envisioned for high-temperature applications[30,31] as presented in Fig. 4e.

**Metal-porogen composite for porous structure development.** Further, the VIPS-3DP approach is applied to print structures with spatially tunable, multi-scale porosity and pore size, which are defined by the inter-filament spacing and intra-filament pores. Such porous features can be controlled by the printing path and the incorporation of a porogen material (such as sodium chloride (NaCl)) respectively, as shown in Fig. 5a. The final intra-filament porosity is easily tunable by controlling the volumetric ratio of porogen within the ink, and such porogen, mixed in the ink preparation step, should not be readily dissolved in the solvent and the nebulized non-solvent environment during printing. The printed parts are deposited in a coagulation and porogen dissolution bath after printing, so the polymer fully solidifies, and the porogen particles are dissolved (Fig. 5b). The dissolved porogen is removed from the deposited filaments due to the presence of interconnected pores from the polymeric phase separation, and the voids left by the dissolution of porogen particles are found in the outermost regions of the filament. NaCl is used as a model porogen due to water being used as the model coagulant and porogen dissolution bath in this study, and no NaCl particles are found in the green body state (Supplementary Fig. 19), confirming the interconnectivity of the entire pore network as water has access to all embedded porogen particles to dissolve them. Upon sintering, the deposited filaments present a network of inter-filament pores as defined by the printing path, and each filament has intra-filament pores, resulting in the multi-scale nature of the pore size found in the printed parts (Fig. 5b, right side). Moreover, VIPS-3DP can be utilized in the framework of fabrication of functionally graded parts since the mixing-and-printing procedure for various inks containing different materials easily enables a spatially varying composition during printing. Structures with a functionally graded composition can be printed with a varying amount of porogen material throughout the print by using a mixer right before printing as shown in the schematic of Fig. 5c, where the amount of porogen is increased from layer to layer. Upon dissolution and sintering, it results in a gradient of porosity along the height of printed parts.

Such spatially tunable, multi-scale porous parts are of great biomedical interest for bone tissue engineering applications due to their higher osseointegration performances[32]. The generated pores enhance the bone-implant anchorage due to the in-growth surrounding bone tissues and properly tune down the mechanical property of implant

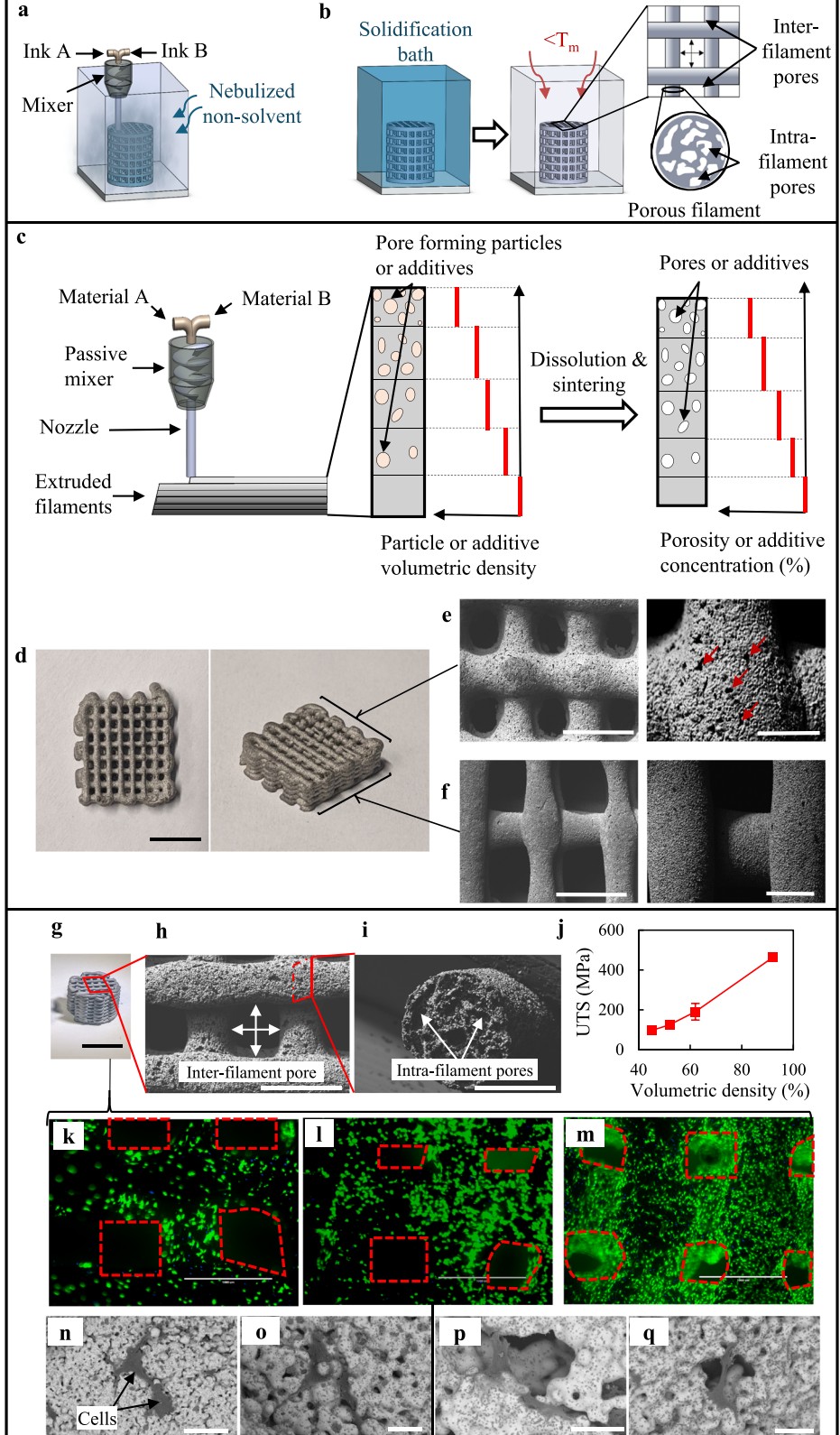

**Fig. 5 | Metal-porogen composite for porous structure development.**
**a** Schematic of VIPS-3DP of parts with multi-scale porosity, **b** post-printing solidification in bath and sintering, **c** printing schematic for functional gradient parts, and **d** printed porous gradient structure with detailed surface morphology for **e** top porous layer and **f** bottom dense layer (Red arrows: open pores located in outer filament region). **g** Porous 316L lattice scaffold, **h** inter-filament pores, and **i** intra-filament pores. **j** UTS as a function of intra-filament volumetric density. **k**–**m** Fluorescent images of cells on the surface of porous scaffolds after culturing for 1, 3, and 7 days, respectively (Green color indicates live cells, and red dashed boxes indicate inter-filament voids). SEM images of MC3T3-E1 cells after 4 days of culture **n**, **o** dense specimens, **p**, **q** porous specimens (Scale bars: 10 mm for (**d**, **g**), 2 mm for (**e**, **f**) (left), 500 μm for (**e**, **f**) (right), 1 mm for (**h**), 500 μm for (**i**), 1 mm for (**k**–**m**), 100 μm for (**n**), 50 μm for (**o**), and 30 μm for (**p**, **q**). Error bars = ± sigma).

parts to avoid undesired stress shielding effects. Pore-induced surface area is advantageous for cellular matter to flow and promote cell attachment, growth, proliferation, and differentiation. More importantly, the use of multi-scale porosity and pore size can result in beneficial bone regeneration dynamics. Generally, smaller pores are better for osteoblast cell attachment, protein absorption, and bone regeneration (osteochondral ossification), while larger pores favor osteogenesis due to the increased space that allows for more vascularization and oxygenation that leads to higher long-term cell proliferation and viability. A lattice structure has been printed for illustration purposes by mixing a 316L powder and ABS-based ink with a NaCl suspension. The results, upon sintering, are shown in Fig. 5d; the top layer exhibits porous features (Fig. 5e, intra-filament porosity ~40%–60%, open pores highlighted by red arrows) while the bottom layer presents a dense morphology (Fig. 5f, intra-filament porosity ~1%–8%). Other stainless-steel lattice scaffolds with a homogeneous composition (Fig. 5g) are also printed for characterization purposes. Further scanning electron microscopy (SEM) images confirm the presence of open pores (pore size found within 20–60 µm) as observed on the outermost surface of the filament (Fig. 5h) and inter-filament pores in the range of 700 µm. The intra-filament pores are also observed in the cross section of the filaments (Fig. 5i), and the average pore size is approximately 120 µm, which matches the porogen size with sintering-induced shrinkage considered and can be easily extendable to other pore sizes by adjusting the size of porogen particles. The presence of pores notably reduces the mechanical properties of the final part; specifically, the UTS is down to 100–200 MPa for representative intra-filament porosity values from approximately 60%–40% (Fig. 5j and Supplementary Fig. 20).

The suitability of VIPS-3DP as a manufacturing route to produce porous biomedical implants is assessed by studying the cytotoxicity of the printed 316L porous parts and the cell proliferation behavior on the 3D-printed implants by using MC3T3-E1 murine preosteoblasts. Cell morphologies are evaluated after culturing for 1, 3, and 7 days. Fluorescent images suggest that the stainless-steel-based structures are not cytotoxic as the cells remain alive throughout 7 days of culture, and the porous nature of the scaffold allows significant cell growth and migration in the porous cavities while being alive. In particular, on Day 1 isolated cells are observed extending and spanning over the porous outer surface (Fig. 5k). Significantly greater cell proliferation is observed on the outer surface on Days 3 and 7 within porous filaments (Fig. 5l, m) when compared to the dense counterparts (Supplementary Fig. 21), proving the good surface chemistry of 316L and confirming the biocompatible features of the porous parts. Additional SEM images show how the cells proliferate on the surfaces of dense (Fig. 5n, o) and porous features (Fig. 5p, q) after 4 days of culture. Interestingly, cellular agglomerates better populate surrounding pore areas where cells bridging over an open pore are seen (Fig. 5p, q), showing that porous features support and better facilitate the cell proliferation process.

## Discussion

In summary, it has been demonstrated that through the use of a versatile vapor-induced phase-separation process, named VIPS-3DP, polymer-based parts can be easily printed, whose solidification mechanism is enabled by the presence of a nebulized non-solvent for polymer. The solvent used can be successfully reclaimed and reused. Such a VIPS-enabled solidification mechanism can be extended to the deposition of polymer-based colloidal inks to obtain composite parts, which enables the fabrication of polymer-metal and polymer-ceramic parts. During composite printing, polymer functions as a binder material, and its binding effect is made possible by VIPS. Most ceramic or metallic particle-based suspensions are not yield-stress fluids, they cannot be printed directly without the use of the VIPS concept for polymer solidification and binding. If a thermal cycle is introduced to debind and remove the polymer from composite parts and induce solid-state sintering, pure metallic, ceramic and composite parts can

be fabricated. In essence, metal printing in this work can be considered as a special case of composite printing where the polymer binder is removed to get metal parts by thermal debinding. Furthermore, parts with spatially gradient and/or tunable multi-scale porosity can be fabricated using VIPS-3DP by using a mixer and combining inter-filament and intra-filament porosities. Such parts may find various applications such as multi-scale porosity bone implants.

Technically speaking, VIPS-3DP provides an alternative DIW approach to printing engineering polymers without heating or tailoring them for certain solidification mechanisms as well as heterogeneous composite structures by using polymer as a binder material. The resulting microstructure is porous due to the nature of phase-separation process, which may be utilized for some porosity-inspired applications but not intended for load-bearing applications directly. For such applications, it should be noted that the pore formation process within the polymeric phase is to be further investigated regarding how the pore size is controlled or affected by the printing process when sacrificial template additives are added during composite printing. In addition, the VIPS-3DP process can be economically implemented by adding a nebulized environment only. Typical DIW processes require the dispensed ink to be solidified in situ to retain the printed form. For VIPS-3DP, the solidification rate is bounded by the diffusion speed of non-solvent vapor, polymer demixing rate during phase separation, non-solvent concentration (RH in this study since water is the non-solvent), and polymer concentration. For a given combination of polymer-solvent-non-solvent, the printing speed is limited by the solidification rate of the dispensed ink, and the printing resolution is limited by the size of the smallest nozzle usable for a given polymeric ink. Furthermore, since the phase-separation process is the mechanism for solidification, the resulting polymer parts may experience post-printing variations, for which should be compensated during the design phase.

## Methods
### Ink preparation

Polymeric ink formulations were prepared by dissolving solid polymers (ABS (ABSplus P430, Stratasys, Eden Prairie, MI, USA) or PAN (150 kDa, Pfaltz and Bauer, Waterbury, CT, USA)) in DMSO (Bioreagent grade, Fisher, Fair Lawn, NJ, USA). ABS-based and PAN-based inks contained a polymeric concentration of 25%–50% and 15%–25%, respectively. The polymeric mixtures were continuously stirred for homogeneous mixing using a roller mixer (DLAB Scientific, Riverside, CA, USA) overnight.

Polymer-metal composites were prepared by mixing 10% (wt%) of copper powders with a mixture of 40% (w/v) ABS and 20% (w/v) TPU (Ellastollan, BASF, Germany) in a 2:1 ratio. Polymer binder solutions for other metal and ceramic composite part printing were prepared by using ABS 25% (w/v) and PAN 15% (w/v) in DMSO, respectively. For 316L printing, metallic suspensions were prepared by mixing the ABS and PAN polymer solutions, respectively, with stainless-steel 316L powders (D10: 4.61 µm, D50: 12.58 µm, D90: 24.44 µm, MSE Supplies, Tucson, AZ, USA). For 316L-HAp metal-ceramic composite printing, 316L and HAp powders (<200 nm, Sigma, St. Louis, MI, USA) were added at a ratio of 95%–5% wt into the ABS solution. For the production of WC-Ni composites, tungsten carbide powders (<2 µm, Sigma, St. Louis, MI, USA) and nickel powders (3–7 µm, Alfa Aesar, Ward Hill, MA, USA) were mixed in a 97–3 wt% ratio into the ABS solution. The powder-to-binder ratios of approximately 55–45 by volume were used for all ink combinations. For the porous structure fabrication, sodium chloride powders (<177 µm, NaCl, GDF Chemicals, Powell, OH, USA) treated by an 80-mesh filter (177 µm in diameter) were mixed at 1:2 and 1:4 weight ratios to 316L powders. ABS solution and powders were then mixed using a centrifugal mixer (AR-100, Thinky, CA, USA) for 2–3 min, and then loaded in a disposable 5 mL syringe fitted with stainless-steel 18–23-gauge tips (330–840 µm in diameter, Nordson EFD, Vilters, Switzerland) for printing.

## Printing

All 3D-printed structures were fabricated using the 5 mL syringes assembled onto a Hyrel Engine SR 3D printer (Hyrel3D, Norcross, GA, USA). A layer thickness of 0.05–0.4 mm and a path speed of 0.5–5.0 mm/s were used. This setup was easily extended to print multi-materials with variable composition by the use of a static mixer (SMH, Hyrel3D, Norcross, GA, USA) while two different inks were simultaneously being fed from two separate printheads. Deionized water mist was supplied simultaneously using a nebulizer (Lumiscope, East Rutherford, NJ, USA) as a non-solvent while printing in order to induce partial solidification onto the structure being printed based on the VIPS mechanism. The 3D-printed green part was then immersed in a water-based coagulation bath (approximately 500 mL) for 1 h to get fully solidified. In this step, the porogen particles, if used, embedded within the intra-filament region were dissolved in the water bath, obtaining a porous green part.

## Antibacterial testing

*Staphylococcus aureus* (*S. aureus*) (ATCC 6358, ATCC, Manassas, Virginia, USA) cells were cultured in 1 mL of trypticase soy broth (TSB) (BD Difco, Becton, New Jersey, USA) in 24-well plates, along with the printed lattice disks. Additional positive controls (without disks), negative controls (without *S. aureus* inoculum), and material controls (non-composite, only polymer-based disks) were used. All groups were incubated at 37 °C, 150 rpm for 24 h. Each test run contained a single specimen (unless specified otherwise), and it was run in triplicate. The antibacterial performance of the printed structures was determined using the Colony Forming Unit (CFU) counting method. For the testing of reused disks, the same sample disks used previously were washed twice with phosphate-buffered saline (PBS, WR, Radnor, Pennsylvania, USA) and once with 75% ethanol prior to the day of the test, then soaked in 75% ethanol for 30 min.

## Debinding and sintering

All 3D-printed samples were heat-treated using a tube furnace (VTF-1700, Zylab, China) with an alumina tube. The parts were placed in magnesia or alumina crucibles. The debinding and sintering cycles were conducted under vacuum conditions obtained by the use of a rotary vane vacuum pump (EQ-FYP, MTI Corp, Richmond, CA, USA). The heating and cooling rates were 5 °C/min, and the thermal cycle involved two separate stages: (i) 1 h at 300 °C and 1 h at 500 °C to avoid thermal cracking of the tube and decompose the polymer, and (ii) a solid-state sintering stage to the 316L and HAp parts at 1300 °C and to the WC-Ni parts at 1450 °C for 2 h. Metallographic samples were obtained by cutting the samples using a diamond saw and mounted in epoxy and polished using standard procedures.

## Sample imaging, porosity, and mechanical properties characterization

The cross sections of polymeric samples were coated with a thin layer of Au-Pd. Imaging was obtained using scanning electron microscopes (Hitachi S3000, Hitachi High-Technologies Co., Tokyo, Japan for non-implant-related applications and Phenom XL, Thermo Fisher Scientific, Waltham, MA, USA for implant-related applications), and the chemical composition was analyzed via energy-dispersive X-ray spectroscopy (Hitachi S3000, Hitachi High-Technologies Co., Tokyo, Japan). Porosity measurements were carried out via image analysis from cross sections using ImageJ (NIH, USA). Flat dog bone specimens were printed under different material conditions, whose dimensions were approximately 30–40 mm long and 8–12 mm wide after sintering. The ultimate tensile strength (UTS) values were measured on a tensile test apparatus (6800 Series, Instron, Norwood, MA, USA) using a crosshead speed of 6–7 mm/min (ASTM E8-03). Three independent trials were conducted under the same conditions.

## Data availability

The data that support the findings of this study are available from the corresponding author upon request.

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

## Acknowledgements

The research was partially supported as part of the AIM for Composites, an Energy Frontier Research Center funded by the U.S. Department of Energy (DOE), Office of Science, Basic Energy Sciences (BES), under award #DE-SC0023389 (Y.H.) (composite printing process development), and by the U.S. National Science Foundation (Grant no. 2315811) (Y.H.) (metal printing process development). In addition, we acknowledge support by the University of Florida (UF) Innovate (Y.H.), the UF Office of Graduate Diversity Initiatives (M.S-G.), and the UF Graduate School (M.S-G.) (VIPS-3DP process development). We also thank Ms. Yunxia Chen, Dr. Kaidong Song, and Mr. Matthew Snyder for their help during the project, Dr. Stephen Miller's group at the UF Chemistry department for the thermogravimetric analysis, and Dr. Nancy J. Ruzycki and Mr. John-Thomas T. Robinson at the UF Materials Science & Engineering department for their support on testing and imaging.

## Author contributions

M.S-G. and Y.H. were responsible for the conception, design, and implementation of the research project. B.R. did literature benchmark research and made substantial contributions to the conceptualization, design, and implementation of the research. B.J.R. was responsible for the polymer printability study and related evaluations. J.G. and G.W. were responsible for the horn antenna design and performance characterization. J.H. and X.J. were responsible for the bacterial testing and result interpretation. W.C. was responsible for the cell viability testing and reporting. B.R., J.Y., G.F., G.W., and X.J. were responsible for the improved implementation of the VIPS-3DP process as well as the review of the manuscript. M.S-G. wrote the first draft and all authors contributed to reviewing and editing.

## Competing interests

M.S-G., B.R., and Y.H. are the authors of three patents that have been granted or filed: US Patent 11,759,999 B2, US Patent 11,833,586 B2, and US Patent Application 63/349,800, describing 3D printing of polymer parts, powder suspension-based parts, and porous and gradient parts using the vapor-induced phase-separation process. The other authors declare no competing interests.
