## [Peer Review File · Nature Communications]

REVIEWER COMMENTS

Reviewer #1 (Remarks to the Author):

see attachment

Reviewer #2 (Remarks to the Author):

This paper introduces a new direct ink approach where the solidification of the solvent-based ink upon deposition is achieved in a “vapor-induced” manner. Namely, The non-solvent is nebulized into the printing environment, creating a near-saturated atmosphere. This facilitates the solvent- non-solvent exchange and rapid solidification in-situ during the printing process, allowing for printing of various material systems. Several of those systems were demonstrated here along with associated practical applications.

This is a well-written, thorough paper, demonstrating several new and exciting applications of direct-ink-writing. There are however several questions regarding the clarity of the method’s novelty. One can deduce that the novelty of this approach compared to the Immersion precipitation approach that was demonstrated earlier lies with the nebulized non-solvent. The claim is that this vapor environment is more tunable than non-solvent bath which allows for control of part properties, mainly the porosity. Yet, this key capability is not really demonstrated. All the examples are printed either in a fully saturated environment or without nebulized non-solvent, which in essence, could be considered as very similar to the two already demonstrated variations of DIW: Immersion precipitation and solvent cast approaches. I believe demonstrations of the process through the tuning of the nebulized environment and thus the part property control that way would be required to truly highlight the innovation here. Without this aspect, most of the key innovation claims does not seem to be dependent on the VIPS approach. Namely, in-situ mixing of inks, manufacturing of multi-material composites etc. can also be realized using the solvent cast or immersion type approaches. The authors are referred to the following work for practical ideas regarding the tuning of the nebulized environment:

Estelle, K. T., & Gozen, B. A. (2021). Humidity-controlled direct ink writing for micro-additive manufacturing with water-based inks. *Journal of Manufacturing Processes*, 69, 583-592.

Reviewer #3 (Remarks to the Author):

This paper presented an interesting 3D printing technique based on vapor-induced phase separation. The authors conducted comprehensive studies and demonstrated that the proposed approach is versatile, and can be used to print parts with polymers, composite, metal, and ceramic. Multi-scale and gradient structures can be printed by using mixer and porogen. The authors also demonstrated various

applications in mechanical, biomedical, and electronic fields. The proposed approach is novel and technically sound. The paper is well-written and organized. The data and analysis are sufficient to support the objective of the work. The reviewer has the following minor comments for the authors to consider and clarify:

(1) The authors mentioned that the phase separation speed depends on the properties of the ink. Specifically, the printing speed for the ABS-based and PAN-based ink is 2-5 mm/s and 0.5-2 mm/s, respectively. This printing speed appears to be significantly slower (approximately 10 times) than that of FDM printers. To provide a more comprehensive understanding of the scalability of the VIPS technique, it would be beneficial if the authors could discuss the potential challenges associated with the slower printing speed. Additionally, could the authors suggest any strategies or future research directions for improving the speed while maintaining the quality of prints?

(2) While the paper presents compelling case for the versatility of the VIPS technique across various materials, it is important to address the advantages of using VIPS for metal printing in comparison to established commercial metal printing technique such as binder jetting. Please provide a more detailed discussion highlighting the unique strengths of VIPS for metal printing, especially regarding factors like cost-effectiveness, material compatibility or potential applications where VIPS outperforms existing methods.

(3) In the VIPS process, the outer region is solidified first. The authors claim that “such a solidification process is intended to present dissolved polymer to ensure adjacent interlayer fusion”. Since the fusion happens at the outer region of the filament, which is solidified, why it can ensure the bonding?

(4) The authors claim that the solvent can be collected. Please provide detailed information on how the solvent in the nebulized non-solvent vapor can be collected. Additionally, it would be valuable to discuss any potential environmental concerns regarding the release of dissolved solvents into the printing environment.

Review for Nature Communications

Title: Vapor-Induced Phase Separation-Enabled Versatile Direct Ink Writing (VIPS-3D)

Authors: Marc Sole-Gras et al.

The authors present a novel approach for extrusion-based 3D printing, which takes advantage of solvent extraction from polymer solution based inks upon extrusion into an environment saturated with a fluid that exhibits a high affinity to the solvent of the ink but does not dissolve the polymer. Extraction of the solvent leads to a solidification of the printed green body. This approach sounds interesting, the manuscript, however, does not provide sufficient information allowing for a scientifically sound evaluation of the concept. As a proof of concept, the authors present small parts (< 5 cm), slowly printed (< 5 mm/s), with poor resolution (nozzle size \approx 0.5 mm). This is far behind the state of the art compared to other DIW techniques. The benefits on the other hand are not clearly highlighted. The manuscript must be far more elaborate with respect to the specifications and limitations of the new concept.

The printed filaments are very non-uniform, particularly around bends and curves, or where filaments of adjacent layers cross. How can this be improved?

What is the shape accuracy, what span can be achieved and how is this affected by the ink composition? For the pure polymer inks it should be thoroughly discussed how polymer molecular weight and concentration, how solvent quality, as well as affinity between solvent and nebulized non-solvent affect printing behavior and strength of the green bodies in a quantitative manner. Essential rheological data (yield stress, viscosity, shear modulus (before and after solvent extraction)) are completely missing.

The authors also show that the new concept can be used to print composite materials including metal and/or ceramic particles. In these cases, the relevance of the VIPS concept is, however, not clear. The flow behavior and strength of such highly filled composite materials is dominated by the properties of the particle network controlled by the (attractive) interactions among the particles. The authors have to show the benefits of extracting the solvent using a nebulized fluid in the printing chamber explicitly. I guess for some of the printed materials there is just no difference using VIPS or not.

Moreover, the authors should be aware that printers with heated stages are commercially available. The authors therefore have to demonstrate the differences between these approaches. Elevated temperature can be used to extract the solvent of the ink, and at a first glance a heating unit seems to be more versatile.

The manuscript includes various chapters describing the performance of printed demonstrators including antennas, respirators, porous tissues for cell growth etc.

These chapters do not provide any new insights about these complex fields of research and can thus be safely omitted. These features of the presented devices are related to the active ingredients and composition of the printed inks, but not to the newly developed VIPS printing concept.

In many aspects these chapters are too superficial and important information is lacking, e.g. how the pore size is controlled or affected by the printing process when NaCl is used as sacrificial template, which as such is nothing new and not directly related to the VIPS concept.

In conclusion, the manuscript is not suitable for publication in its present form.
A fundamental revision is mandatory which has to be considered as new submission.

List of Changes / Responses Letter

Article ID: NCOMMS-23-03487

Title: Vapor-Induced Phase Separation-Enabled Versatile Direct Ink Writing (VIPS-3D)

Changes/Responses per the comments from Referee 1

We highly appreciate the encouragement and comments on our study and have updated our manuscript as suggested. In particular, all major changes are highlighted in red ink in the revised manuscript.

1. The authors present a novel approach for extrusion-based 3D printing, which takes advantage of solvent extraction from polymer solution based inks upon extrusion into an environment saturated with a fluid that exhibits a high affinity to the solvent of the ink but does not dissolve the polymer. Extraction of the solvent leads to a solidification of the printed green body. This approach sounds interesting, the manuscript, however, does not provide sufficient information allowing for a scientifically sound evaluation of the concept. As a proof of concept, the authors present small parts (< 5 cm), slowly printed (< 5 mm/s), with poor resolution (nozzle size \approx 0.5 mm). This is far behind the state of the art compared to other DIW techniques. The benefits on the other hand are not clearly highlighted. The manuscript must be far more elaborate with respect to the specifications and limitations of the new concept.

Change(s): Thank you very much for your encouraging and constructive comments on this new VIPS-3DP concept. While it is interesting and novel, we understand that much still needs to be studied for it to be a mature technology.

The proposed process is not aimed at replacing other DIW processes. Instead, it provides an alternative DIW approach to printing engineering polymers without heating or tailoring them for certain solidification mechanisms; it also enables the fabrication of heterogenous composite structures by using polymer as a binder material. The resulting microstructure is porous due to the nature of phase-separation process, making this technology suitable for some porosity-inspired applications. In addition, the VIPS-3DP process can be economically implemented by simply adding a nebulized environment to a DIW equipment.

As recommended, a deeper evaluation has been conducted to study the printability, achievable printing resolution, print shape accuracy, part microstructure, and effects of material properties and printing conditions (i.e., printing speed, relative humidity (RH), and nozzle size). This aims at gaining valuable insights on the ink printability, part microstructure, and polymeric green body strength during printing a pure polymer ABS-based ink. Most of the results are presented in addressing your second comment, with some included in both the revised manuscript and supplemental information.

While the part size is limited by our printer, the printing/path speed and printing resolution do have their own limits due to the VIPS-based solidification physics. Once ink is dispensed out of the nozzle, it needs to be solidified *in situ* within a specific timeframe to retain its shape and ensure proper bonding. For such a solidification dynamics analysis, the solidification timescale is estimated via optical measurements. Figure R1 presents an example, where it is shown a typical

20% (w/v) ABS-based ink being printed on a glass substrate and observed (bottom view) over time to quantify the solidification rate during the VIPS printing process by capturing the opacification of the filament. For the 20% (w/v) ink under 40% RH, it takes approximately 90 seconds for the opacification to be fully completed. The solidification front travels inwards (white arrows, inset). The printing speed needs to match the solidification rate for a good shape fidelity and proper inter-layer fusion. For the 20% (w/v) ink under higher RH, the opacification is a much more rapid process (<10 seconds as this is the time it takes for the contrast to adjust with our imaging setup). It should be pointed out that higher concentration ABS inks present significantly more opacity in the liquid phase, inhibiting any meaningful observation of the change in opacity over time. The information is included as S3 in the revised SI file, and the manuscript has been updated on Page 5 as follows:

“... This leads to slower demixing kinetics of PAN upon exposure to the non-solvent (water) as seen from the different polymer-solvent interaction parameters g_{23} as discussed in SI S2.2. **Typical evolution of the solidification front after filament printing can be seen in SI S3. ...**”

Figure R1. Evolution of the solidification front of 20% (w/v) ABS ink at 40% RH over time showing 0, 30, 60, 75, and 90 seconds after filament printing. Scales bars: 2 mm.

In general, this solidification rate is bounded by the diffusion speed of non-solvent vapor, polymer demixing rate during phase separation, non-solvent concentration (vapor-induced RH in this study since water is the non-solvent), and polymer concentration. For a given combination of polymer-solvent-non-solvent, the printing speed is limited by the solidification rate of the dispensed ink, and the printing resolution is limited by the size of the smallest nozzle usable for a given polymeric ink as most dissolved engineering polymers are only slightly shear-thinning. Furthermore, since the phase separation process is the mechanism for solidification, the resulting polymer parts may experience post-printing variations under certain circumstances, for which should be compensated during the design phase.

That being said, we have conducted additional experiments to investigate the achievable resolution per our material system (ABS, DMSO, and water). The achievable printing resolution is affected

by the RH level, nozzle diameter, and printing speed. The experiments under various RH levels and nozzle diameters show that a higher RH level leads to a smaller filament diameter due to the faster solidification and depressed ink spreading (Figure R2(a)). At RH = 99%, the solidified filament diameter of 0.89 mm and 1.50 mm are achieved using 0.51 mm and 1.36 mm nozzles, respectively (Figure R2(b)). By using an even smaller nozzle (diameter = 0.10 mm) combined with a higher printing speed (5 mm/s), filaments with a 0.29 mm resolution are achieved (Figure R2(c)). For the demonstration of high-resolution printing using VIPS-3DP, a miniature lattice is printed as shown in Figure R2(d), and each segment is approximately 0.30 mm tall (three layers for 0.10 mm each) and 0.30 mm wide (two filaments).

Figure R2. (a) Schematic of filament formation under different humidity levels. (b) Effects of humidity levels (40%-99%) on the printed 40% (w/v) ABS filament diameter using gauge 15 (diameter of 1.36 mm) and gauge 25 (diameter of 0.51 mm) nozzles. (c) Effects of printing speed (1.5 – 5.0 mm/s) on the printed 30% (w/v) ABS filament diameters using gauge 20 (diameter of 0.10 mm) at 99% RH. (d) A miniature lattice printed using 35% (w/v) ABS under 99% RH with a gauge 32 (diameter of 0.10 mm) nozzle. Scale bars are 0.50 mm if not specified.

As suggested, the specification and limitations of VIPS-3DP are included in the Summary and Perspective section (renamed from the Summary section) on Page 15 as follows:

“... Technically speaking, VIPS-3DP provides an alternative DIW approach to printing engineering polymers without heating or tailoring them for certain solidification mechanisms as well as heterogenous composite structures by using polymer as a binder material. The resulting microstructure is porous due to the nature of phase-separation process, which may be utilized for some porosity-inspired applications. For such applications, it should be noted that the pore formation process within the polymeric phase is to be further investigated regarding how the pore size is controlled or affected by the printing process when sacrificial template additives are added during composite printing. In addition, the VIPS-3DP process can be economically implemented by adding a nebulized environment only. Typical DIW processes require the dispensed ink to be solidified *in situ* to retain the printed form. For VIPS-3DP, the solidification rate is bounded by the diffusion speed of non-solvent vapor, polymer demixing rate during phase separation, non-solvent concentration (RH in this study since water is the non-solvent), and polymer concentration. For a given combination of polymer-solvent-non-solvent, the printing speed is limited by the solidification rate of the dispensed ink, and the printing resolution is limited by the size of the smallest nozzle usable for a given polymeric ink as most dissolved engineering polymers are only slightly shear thinning. Furthermore, since the phase separation process is the mechanism for solidification, the resulting polymer parts may experience post-printing variations, for which should be compensated during the design phase.”

2. The printed filaments are very non-uniform, particularly around bends and curves, or where filaments of adjacent layers cross. How can this be improved? What is the shape accuracy, what span can be achieved and how is this affected by the ink composition? For the pure polymer inks it should be thoroughly discussed how polymer molecular weight and concentration, how solvent quality, as well as affinity between solvent and nebulized non-solvent affect printing behavior and strength of the green bodies in a quantitative manner. Essential rheological data (yield stress, viscosity, shear modulus (before and after solvent extraction)) are completely missing.

Change(s): As suggested, we have further conducted studies regarding the printability, print shape accuracy, and effects of material properties (polymer concentration) and printing conditions (RH in addition to the printing speed and nozzle size) on the ink printability, part microstructure, and polymeric green body strength during printing a pure polymer ABS. Since the molecular weight information of the commercially available ABS material is not available, we have only attempted to vary the ABS concentration, which may have a similar effect on the printing performance during VIPS-3DP. We have also investigated the effects of polymer materials with different chemical affinities between solvent (DMSO) and nebulized non-solvent (water) on the post-printing solidification process and final microstructures. As suggested, essential rheological data has been acquired and presented herein. The maximum filament span is not explored due to our hardware limitation and will be investigated in a future study.

2a) Ink printability

First, the effects of printing conditions (RH) and material properties (ABS concentration) on the filament diameter have been further evaluated during printing pure ABS (Figure R3). For a given

ABS ink, the printing behavior is highly related to the increased RH from ink spreading to well-defined filament to deformed filament as exemplified in Figure R3(a). At low RH levels, the ink is not solidified quickly enough, and thus, there is enough time for the ink to deform due to its own weight and the lack of sufficient surface tension and hardening degree. This effect is most apparent for the low viscosity ink compositions such as 20% (w/v) ABS since these samples have low shear moduli resulting in poor stability. On the other hand, higher RH levels may induce premature solidification-induced defects such as wavy filament shapes due to the fast solidification dynamics in place.

Results show that 20%-60% (w/v) ABS inks under 40% and 70% RH have a larger filament diameter than that of the nozzle (herein a gauge 22 nozzle with an inner diameter of 0.41 mm) as the insufficient solidification after extrusion results in poor shape fidelity, as seen in Figure R3(b). In particular, the 20% (w/v) ABS ink shows the most serious ink spreading due to the low viscosity. However, the 40% and 60% (w/v) ABS inks under 99% RH show well-defined filaments of diameter close to that of the nozzle with minimal filament dispersion, which is attributed to the high solidification rate at the highest RH level. It is noted that the 20% (w/v) ABS ink under 99% RH may be solidified with a wavy filament geometry resulting from rapid solidification-induced uneven deformation over relatively low strength filaments (due to the low polymer concentration).

Figure R3. (a) Schematics of different printed filament morphologies with increased RH and (b) effect of ABS concentration (20%-60% w/v) on the filament diameter under 40%, 70%, and 99% RH. Inset: images of the solidified filaments. Dashed lines in the inset: nozzle diameter. (Scale bars: 2 mm)

For a detailed printability study during VIPS-3DP, a printability factor ($Pr = \frac{L^2}{16A}$) is used to quantitatively evaluate the filament fidelity using a lattice design [Soltan 2019], shown in Figure R4(a) and (b), where L is the perimeter length of a single cell of the lattice and A is the measured blank area enclosed by the printed strands. Pr of 1 is the ideal value, while Pr of less than 1 results in a smaller perimeter to area ratio than that is present in a square geometry. A value greater than 1 also indicates deformed filaments with an increased cell perimeter. Images of printed lattice structures of 20%-30% (w/v) ABS under 40% RH (0.10 mm nozzle at 5 mm/s) are shown in Figure R4(c). It shows that the 30% (w/v) ABS ink has the best shape fidelity, while 20% (w/v) ABS filaments spread and fuse together due to the low viscosity and slow solidification. The measured Pr values are present in Figure R4(d), which are consistent with the print images. The Pr value is also affected by the RH conditions as seen from R4(e) and (f). For the 25% (w/v) ABS ink being

printed at 3 mm/s using a 0.10 mm nozzle, the 70% RH condition results in the best printability factor around 1. The ink cannot have a good printability factor under 40% RH even when the printing speed is reduced from 5 mm/s to 3 mm/s in order to let the ink have sufficient time to solidify and retain a good shape fidelity. Even though printing under RH values of 99% may theoretically yield Pr values close to 1, it must be noted that such conditions are not advisable as the ink may solidify too fast and clog and block the dispensing nozzle, resulting in deformed filaments.

Figure R4. Quantitative evaluation of printing fidelity. (a) Lattice design for printing evaluation. (b) Schematic of printability factor (Pr) measurement. (c) Printing results of 20-30% (w/v) ABS inks under 40% RH using a 0.10 mm nozzle at 5 mm/s. (d) Pr results for (c). (e) Printing results of 25% (w/v) ABS ink under 40%, 70%, and 99% RH using a 0.10 mm nozzle at 3 mm/s. (f) Pr results for (e) (scale bars: 5 mm, and N/A: not applicable).

In terms of the print shape accuracy, resulting polymer parts may experience some post-printing shrinkage, given that solvent is extracted out of the deposited filament as a result of the phase separation process. The level of shrinkage depends on the RH value (Figure R5(a) and (b)), polymer concentration (Figure R5(a) and (c)), and polymer material (SI Figure S4.3). To study the effect of RH on the shape accuracy, a circular structure with an inner diameter (ID) of 10.0 mm, an out diameter (OD) of 10.9 mm, and a wall thickness of 0.45 mm (Figure R5(a)) is designed and printed to evaluate the print shape accuracy using a gauge 22 nozzle (diameter of 0.41 mm) at 4 mm/s. The 40% (w/v) ABS ink was printed under the 40%, 70%, and 99% RH levels, and the

printing results are shown in Figure R5(b). The printed circles are smooth with good shape fidelity, and the measured ID, OD, and wall thickness are close to the design values (Figure R5(d)). Of them, the 40% RH level results in the best print shape accuracy with slight shrinkage.

To study the effect of polymer concentration on the shape accuracy, a circular structure was similarly printed while using the 20%, 30%, and 40% (w/v) ABS inks using a gauge 22 nozzle (diameter of 0.41 mm) at 4 mm/s under 40% RH (Figure R5(c)). The 20% ABS ink has severe spreading due to the low polymeric concentration, resulting in a higher OD and a lower ID than design values, while the 30% and 40% (w/v) ABS inks result in relatively accurate prints due to the combined effect of ink viscosity (for spreading) and solidification speed (Figure R5(e)). Of them, the 40% (w/v) ABS ink results in the best print shape accuracy with slight deviation.

As observed, the good combination of RH and polymer concentration is 70% RH and 30-40% (w/v) ABS during VIPS-3DP ABS printing.

Figure R5. Post-printing solidification-induced filament diameter variation as a function of RH and polymer concentration. (a) Shape design, (b) printing results of 40% (w/v) ABS under 40%, 70%, and 99% RH levels, (c) printing results of 20%, 30%, and 40% (w/v) ABS under 40% RH, and (d-e) measurements of the inner diameter (ID), out diameter (OD), and wall thickness of (b-c), with comparison to the design values (dashed lines).

To study the effect of polymer material on the shape accuracy, both ABS and polyacrylonitrile (PAN) were used as two example polymer materials with different chemical affinities with the solvent (DMSO). Because PAN has a better affinity due to a lower relative energy difference (RED) value than ABS (Supplementary Information Table 1), it leads to a slower phase-separation process. The slow solidification rate of PAN allows it to shrink as it gets slowly solidified and thus, presenting a high post-printing shrinkage degree and smaller pore sizes. In contrast, ABS solidifies fast due to the rapid solvent extraction, having a low post-printing shrinkage degree and forming highly porous morphology with large pore size. An example of the as-printed and as-solidified parts, 20% (w/v) PAN-based semicircles before and after solidification are shown in Figure R6(a) and (b), presenting a clear reduction in diameter as indicated by black arrows. For comparison in terms of the print shape accuracy, the shrinkage magnitude of the 20% (w/v) ABS and 20% (w/v) PAN inks is quantified to be approximately 3% and 30%, respectively, as seen in Figure R5(b), showing that polymers having a low RED value may shrink more significantly.

Figure R6. (a) Tube printing using 20% (w/v) PAN (scale bar = 10 mm) and (b) shrinkage measurements for ABS and PAN.

By leveraging the printing knowledge mentioned above, we have printed additional structures with improved quality (Figure R7).

Figure R7. VIPS printing demonstrations showing some printed structures with different features and filament diameters. Top left spiral printed using 35% (w/v) ABS with a gauge 32 (diameter of 0.10 mm) under 40% RH. Top middle left bicycle printed using 35% (w/v) ABS with a gauge 32 under 70% RH. Top middle right and top right vases both printed using 35% (w/v) ABS with a gauge 32 under 40% RH. Middle left frog printed using 35% (w/v) ABS with a gauge 27 (diameter of 0.20 mm) under 40% RH. Middle 2D lattice printed using 35% (w/v) ABS with a gauge 27 under 99% RH. Middle right vase printed using 35% (w/v) ABS with a gauge 32 under 70% RH. Bottom 3D lattice printed using 35% (w/v) ABS with a gauge 32 under 99% RH. Insets: structure design. Scale bars: 5 mm (if not specified).

The above information is selectively included as S4 in the revised SI file.

2b) Part microstructure

In addition, the effects of ABS concentrations under different RH levels on the microstructures have also been investigated, and the results are shown in Figure R8. The polymer ABS is chosen for investigations for its proneness to printing-induced porosity when compared to PAN. All the samples show a similar pore distribution, where a dense skin layer is observed on the surface, micropores in the intermediate region, and macropores in the innermost region (SI Figure S5.3). The polymer concentration plays a role in the final porous microstructure formation, and this relationship is well-established [Mulder1996] [Chang2019]. For higher-concentration polymer inks, less amount of solvent needs to be extracted, and this results in a theoretically decreased pore size. The finger-like porous structures are oriented along the solidification direction with distributed porous layers and lamellar dense interlayers. No significant morphological difference is found among different RH levels. However, under a higher RH level, the porous and layers are not easily differentiable due to the faster demixing kinetics.

Figure R8. Morphology of porous structures of ABS with different concentrations and RH levels in the printing environment (Arrows: solidification direction along the layer thickness. Scale bars: 80 μm).

2c) Polymeric green body strength

The strength of the green bodies has been studied during printing the 40% (w/v) ABS ink under different RH levels (40%, 70%, and 99%) using a gauge 22 (diameter of 0.41 mm) nozzle at 3 mm/s. A dog bone model with a gauge cross section of approximately 1.35 mm (3 filaments in width) and 1.00 mm (4 layers in height) is used for the tensile test [Jin2016] on an eXpert 4000 MicroTester testing system (Admet, Norwood, MA). A stretching speed of 0.05 mm/sec was applied while recording the load on a 1000 g load cell. The low RH allows for the extruded filament to flow and remain liquid for many seconds, resulting in a much higher degree of compositional homogeneity within the printed part and therefore a higher ultimate tensile strength (Figure R9). During this liquid stage prior to solidification, the filaments fuse into one instead of keeping the structure of some connected filament paths. As observed, the 40% RH sample also fractures with a consistent and flat fracture surface perpendicular to the axial strain due to its good ductility. In comparison, the samples fabricated under the 70% and 99% RH are more brittle due to larger internal pores (higher amount of stress concentration regions) and other fabrication-induced defects such as the premature-solidification-induced deformed or broken patterns. The interlayer adhesion under high RH levels is also reduced compared to that under the 40% RH, resulting in a reduced mechanical strength. In particular, premature solidification of the extruded filament under

the 99% RH introduces more printing defects, further reducing the mechanical strength and creating a greater degree of strength variability. It is noted that the ABS green body strength is much lower than that of typical ABS products due to the porous microstructure-induced weakening effect. As such, polymeric-based VIPS-3DP printed green bodies are not good for load-bearing applications.

Figure R9. Mechanical strength of 40% (w/v) ABS under 40%, 70%, and 99% RH.

As suggested by the reviewer, rheological test has been conducted as described in a previous study [Ren2023], and the results are shown in Figure R10. While some inks may show minor shear-thinning behavior, the results do not show a clear yield-stress property of any of our inks (polymeric dissolved solution or colloidal suspension), meaning that these inks cannot be easily printed using a conventional DIW approach.

Figure R10. Rheological properties of various polymer-based inks (left: viscosity, and right: shear moduli).

The manuscript has been updated accordingly as follows:

on Page 5:

“One of the key advantages of using VIPS as a solidification mechanism is that such solidification dynamics can be adjusted by controlling the delivery of non-solvent to the printing area, **specifically, the RH level herein**. Printing under room conditions results in overall slower phase-separation dynamics when compared to that under a nebulized printing zone, as the amount of water in the environment is significantly different (RH: 40% vs 99%). **For a given polymeric ink, the printing behavior is highly related to the increased RH from ink spreading to well-defined filament to deformed filament as exemplified in Fig. 2(b1)**. At low RH levels, the ink is not solidified quickly enough, and thus, there is enough time for the ink to deform due to its own weight and the lack of sufficient surface tension and hardening degree. This effect is most apparent for the low viscosity ink compositions such as 20% (w/v) ABS since these samples have low shear moduli resulting in poor stability. On the other hand, higher levels of RH may induce premature solidification-induced defects such as wavy filament shapes due to the fast solidification dynamics in place. Results show that 20%-60% (w/v) ABS inks under 40% and 70% RH have a larger filament diameter than that of the nozzle (herein a gauge 22 nozzle with an inner diameter of 0.41 mm) as the insufficient solidification after extrusion results in poor shape fidelity, as seen in Fig. 2(b2). **More detailed printability and print shape accuracy studies can be found in SI S4**. As observed, the good combination of RH and polymer concentration is 70% RH and 30-40% (w/v) ABS during VIPS-3DP ABS printing. For the demonstration of high-resolution printing using VIPS-3DP, a miniature ABS lattice is printed as shown in Fig. 2(c1), and each segment is approximately 0.30 mm tall (three layers for 0.10 mm each) and 0.30 mm wide (two filaments).”

and Page 15:

“... Furthermore, since the phase separation process is the mechanism for solidification, the resulting polymer parts may experience post-printing variations, for which should be compensated during the design phase.”

In addition, Figure 2 has been updated as follows:

Figure 2. ABS-based printing results: (a) sequence of a vase-like structure being printed, (b1) schematics of different printed filament morphologies, (b2) effects of ABS polymer ink concentrations (20%-60% w/v) on the printed filament diameters under 40%, 70%, and 99% RH (Inset: images of the solidified filaments, and dashed lines in the inset: nozzle diameter. Scale bars: 2 mm), (c1) a VIPS-3DP printed lattice part, (c2)-(c3) morphology details of polymeric cross sections, and (d) overhang filament printing process. (Scale bars in (a) and (d): 10 mm).

3. The authors also show that the new concept can be used to print composite materials including metal and/or ceramic particles. In these cases, the relevance of the VIPS concept is, however, not clear. The flow behavior and strength of such highly filled composite materials is dominated by the properties of the particle network controlled by the (attractive) interactions among the

particles. The authors have to show the benefits of extracting the solvent using a nebulized fluid in the printing chamber explicitly. I guess for some of the printed materials there is just no difference using VIPS or not.

Change(s): As suggested by the reviewer, we have tried to make the relevance of the VIPS-3DP concept to composite printing clearer by adding additional comments. Such a VIPS-enabled solidification mechanism can be extended to the deposition of polymer-based colloidal inks to obtain composite parts, which enables the fabrication of polymer-metal and polymer-ceramic parts. During composite printing, polymer functions as a binder material, and its binding effect is made possible by VIPS. From this perspective, metal printing in this study can be considered as a special case of composite printing where the polymer binder is removed to get metal parts by thermal debinding.

While soft polymeric microparticles, which may have yield-stress fluid properties due to the attractive or repulsive interaction among the particles, have been utilized for various 3D printing applications, such an interaction is not common for ceramic or metallic particle-based suspensions as seen from Figure R11 (stainless steel powder (316L)-based ABS suspension as an example). Without the use of VIPS for polymer solidification and binding, the composite printing examples as presented in this study are not feasible.

Figure R11. Rheological measurements of a polymer-stainless steel particle ink (left: viscosity, and right: shear moduli).

The manuscript has been updated on Page 15 to reflect the above discussion:

“..., which enables the fabrication of polymer-metal and polymer-ceramic parts. **During composite printing, polymer functions as a binder material, and its binding effect is made possible by VIPS. Most ceramic or metallic particle-based suspensions are not yield-stress fluids, they cannot be printed directly without the use of the VIPS concept for polymer solidification and binding.** If a thermal cycle is introduced to debind and remove the polymer from composite parts and induce solid-state sintering, pure metallic, ceramic and composite parts can be fabricated. **In essence, metal printing in this work**

can be considered as a special case of composite printing where the polymer binder is removed to get metal parts by thermal debinding. ...”

4. Moreover, the authors should be aware that printers with heated stages are commercially available. The authors therefore have to demonstrate the differences between these approaches. Elevated temperature can be used to extract the solvent of the ink, and at a first glance a heating unit seems to be more versatile.

Change(s): We agree that there are commercially available printers with heated stages that may be utilized for 3D printing. The reasons that VIPS is utilized instead of high-temperature evaporation in this study are based on the following considerations. First, environmentally conscious manufacturing is a main consideration in this study. One of advantage of relying on low vapor pressure solvents (such as DMSO herein) for the VIPS-3DP purposes is the potential reclamation of solvents used, which results in a lower environmental footprint and lower solvent-based fumes exposure to users and equipment when compared to other evaporation-based DIW processes. In this study, DMSO used has been successfully reclaimed post-printing via a simple distillation setup, significantly reducing the footprint of the presented 3D printing technology. Second, the high temperature-induced thermal gradient may lead to undesirable material degradation, manifesting issues like warping and other defects during deposition. Third, the temperature field away from heated stage is difficult to be controlled for effective phase separation and solidification during printing tall structures.

For better clarification, the manuscript has been updated on Page 4 as follows:

“The model solvent used herein is dimethyl sulfoxide (DMSO) because of its nontoxicity, nonvolatility due to its low vapor pressure and easiness to be separated from the water used herein as the non-solvent, via a distillation process. **From an environmental perspective, the use of low-volatility solvents enables their reclamation after the printing process, significantly reducing the environmental footprint as well as the fume exposure to users and equipment during printing when compared to other evaporation-based phase separation processes. ...”**

5. The manuscript includes various chapters describing the performance of printed demonstrators including antennas, respirators, porous tissues for cell growth etc. These chapters do not provide any new insights about these complex fields of research and can thus be safely omitted. These features of the presented devices are related to the active ingredients and composition of the printed inks, but not to the newly developed VIPS printing concept. In many aspects these chapters are too superficial and important information is lacking, e.g. how the pore size is controlled or affected by the printing process when NaCl is used as sacrificial template, which as such is nothing new and not directly related to the VIPS concept.

Change(s): As pointed out by the reviewer, the features of the presented devices are related to the active ingredients and composition of the printed inks, but not to the newly developed VIPS-3DP printing concept. We agree that VIPS-3DP merely helps print the inks into certain 3D structures for various applications, which is the aim of this study, to demonstrate that VIPS is a versatile printing process delivering a wide range of material-related outputs. We agree that better

applications can be explored to show the great potential of the VIPS-3DP concept by collaborating further with research communities to explore the use of the proposed technology to the pursue of novel areas. Since the other two reviewers appreciate these applications, we have added an in-depth evaluation of the printing dynamics of polymer-based inks during VIPS-3DP printing as suggested while keeping such applications.

Regarding composite printing, the reviewer has raised a wonderful point on the printing physics such as how the pore size is controlled or affected by the printing process when sacrificial template additives are added. As recognized by the reviewer, this is not related to the VIPS-3DP concept, and we will investigate the physics during composite printing in a future study. Herein, we merely present some printing results to demonstrate the 3D printing-related enabling capability of the VIPS concept.

For better clarification, the manuscript has been updated on Page 15 as follows:

“... Technically speaking, VIPS-3DP provides an alternative DIW approach to printing engineering polymers without heating or tailoring them for certain solidification mechanisms as well as heterogenous composite structures by using polymer as a binder material. The resulting microstructure is porous due to the nature of phase-separation process, which may be utilized for some porosity-inspired applications. **For such applications, it should be noted that the pore formation process within the polymeric phase is to be further investigated regarding how the pore size is controlled or affected by the printing process when sacrificial template additives are added during composite printing.** In addition, ...”

Changes/Responses per the comments from Referee 2

We highly appreciate the encouragement and comments on our study and have updated our manuscript as suggested. In particular, all major changes are highlighted in red ink in the revised manuscript.

1. This is a well-written, thorough paper, demonstrating several new and exciting applications of direct-ink-writing. There are however several questions regarding the clarity of the method's novelty. One can deduce that the novelty of this approach compared to the Immersion precipitation approach that was demonstrated earlier lies with the nebulized non-solvent. The claim is that this vapor environment is more tunable than non-solvent bath which allows for control of part properties, mainly the porosity. Yet, this key capability is not really --demonstrated. All the examples are printed either in a fully saturated environment or without nebulized non-solvent, which in essence, could be considered as very similar to the two already demonstrated variations of DIW: Immersion precipitation and solvent cast approaches. I believe demonstrations of the process through the tuning of the nebulized environment and thus the part property control that way would be required to truly highlight the innovation here. Without this aspect, most of the key innovation claims does not seem to be dependent on the VIPS approach. Namely, in-situ mixing of inks, manufacturing of multi-material composites etc. can also be realized using the solvent cast or immersion type approaches.

Change(s): Thanks to the constructive comment, we agree that the demonstrations of the printing process through the tuning of the nebulized environment or relative humidity (RH) and thus the part property control would highlight our innovation better. In general, the ink solidification rate during VIPS-3DP is bounded by the diffusion speed of non-solvent vapor, polymer demixing rate during phase separation, non-solvent concentration (RH in this study), and polymer concentration. Of them, the tuning of RH for 3D printing makes the VIPS-3DP process different from the immersion precipitation and solvent cast approaches. As such, we have designed and fabricated a tunable nebulized environment as recommended and further utilized it to study the effects of nebulized relative humidity (RH) on the ink printability, printing resolution, shape accuracy, part strength, and part microstructure. The related results are presented as follows.

1a) Effect of relative humidity on the printability and printing resolution

Once ink is dispensed out of the nozzle, it needs to be solidified *in situ* to retain its shape within a given timescale to ensure proper bonding is ensured. For such a solidification dynamics analysis, the solidification timescale is estimated via optical measurements, and Figure R1 presents an example, where it is shown a typical 20% (w/v) ABS-based ink being printed on a glass substrate and observed (bottom view) over time to quantify the solidification rate during the VIPS printing process by capturing the opacification of the filament. For the 20% (w/v) ink under 40% RH, it takes approximately 90 seconds for the opacification to be fully completed. The printing speed needs to match the solidification rate for a good shape fidelity and proper inter-layer fusion. For the 20% ink under higher RH, the opacification is a much more rapid process (<10 seconds as this is the time it takes for the contrast to adjust with our imaging setup). It should be pointed out that higher concentration ABS inks present significantly more opacity in the liquid phase, inhibiting any meaningful observation of the change in opacity over time. The information is included as S3 in the revised SI file, and the manuscript has been updated on Page 5 as follows:

“... This leads to slower demixing kinetics of PAN upon exposure to the non-solvent (water) as seen from the different polymer-solvent interaction parameters g_{23} as discussed in SI S2.2. **Typical evolution of the solidification front after filament printing can be seen in SI S3. ...**”

Figure R1. Evolution of the solidification front of 20% (w/v) ABS ink at 40% RH over time showing 0, 30, 60, 75, and 90 seconds after filament printing. Scales bars: 2 mm.

In general, this solidification rate is bounded by the diffusion speed of non-solvent vapor, polymer demixing rate during phase separation, non-solvent concentration (relative humidity (RH) are more brittle due to larger internal pores), and polymer concentration. For a given combination of polymer-solvent-non-solvent, the printing speed is limited by the solidification rate of the dispensed ink, and the printing resolution is limited by the size of the smallest nozzle usable for a given polymeric ink as most dissolved engineering polymers are only slightly shear thinning. Furthermore, since the phase separation process is the mechanism for solidification, the resulting polymer parts may experience post-printing variations under certain circumstances, for which should be compensated during the design phase.

That being said, we have conducted additional experiments to investigate the achievable resolution per our material system (ABS, DMSO, and water). The achievable printing resolution is affected by the nozzle diameter, printing speed, and RH level. The experiments under various RH levels and nozzle diameters show that a higher RH level leads to a smaller filament diameter due to the faster solidification and depressed ink spreading (Figure R2(a)). At RH = 99%, the solidified filament diameter of 0.89 mm and 1.50 mm are achieved using 0.51 mm and 1.36 mm nozzles, respectively (Figure R2(b)). By using an even smaller nozzle (diameter = 0.10 mm) combined with a higher printing speed (5 mm/s), filaments with a 0.29 mm resolution are achieved (Figure R2(c)). For the demonstration of high-resolution printing using VIPS-3DP, a miniature lattice is printed as shown in Figure R2(d), and each segment is approximately 0.30 mm tall (three layers for 0.10 mm each) and 0.30 mm wide (two filaments).

Figure R2. (a) Schematic of filament formation under different humidity levels. (b) Effects of humidity levels (40%-99%) on the printed 40% (w/v) ABS filament diameter using gauge 15 (diameter of 1.36 mm) and gauge 25 (diameter of 0.51 mm) nozzles. (c) Effects of printing speed (1.5 – 5.0 mm/s) on the printed 30% (w/v) ABS filament diameters using gauge 20 (diameter of 0.10 mm) at 99% RH. (d) A miniature lattice printed using 35% (w/v) ABS under 99% RH with a gauge 32 (diameter of 0.10 mm) nozzle. Scale bars are 0.50 mm if not specified.

Furthermore, the effects of printing conditions (RH) and material properties (ABS concentration) on the filament diameter have been further evaluated during printing pure ABS (Figure R3). For a given ABS ink, the printing behavior is highly related to the increased RH from ink spreading to well-defined filament to deformed filament as exemplified in Figure R3(a). At low RH levels, the ink is not solidified quickly enough, and thus, there is enough time for the ink to deform due to its own weight and the lack of sufficient surface tension and hardening degree. This effect is most apparent for the low viscosity ink compositions such as 20% (w/v) ABS since these samples have low shear moduli resulting in poor stability. On the other hand, higher levels of RH may induce premature solidification-induced defects such as wavy filament shapes due to the fast solidification dynamics in place.

Results show that 20%-60% (w/v) ABS inks under 40% and 70% RH have a larger filament diameter than that of the nozzle (herein a gauge 22 nozzle with a diameter of 0.43 mm) as the insufficient solidification after extrusion results in poor shape fidelity, as seen in Figure R3(b). In particular, the 20% (w/v) ABS ink shows the most serious ink spreading due to the low viscosity. The 40% and 60% (w/v) ABS inks also have the filament diameter increase under 40% RH but show well-defined filaments of diameter close to that of the nozzle under 99% RH with minimal filament dispersion, which is attribute to the high solidification rate at the highest RH level. It is noted that the 20% (w/v) ABS ink under 99% RH may be solidified with a wavy filament geometry resulting from rapid solidification-induced uneven deformation over relatively low strength filaments (due to the low polymer concentration).

Figure R3. (a) Schematics of different printed filament morphologies with increased RH and (b) effect of ABS concentration (20%-60% w/v) on the filament diameter under 40%, 70%, and 99% RH. Inset: images of the solidified filaments. Dashed lines in the inset: nozzle diameter. (Scale bars: 2 mm)

For a detailed printability study during VIPS-3DP, a printability factor ($Pr = \frac{L^2}{16A}$) is used to quantitatively evaluate the filament fidelity using a lattice design [Soltan 2019], shown in Figure R4(a) and (b), where L is the perimeter length of a single cell of the lattice and A is the measured blank area enclosed by the printed strands. Printability factor of 1 is the ideal value. Printability factor of less than 1 results in a smaller perimeter to area ratio than that is present in a square geometry. A value greater than 1 also indicates deformed filaments with an increased cell perimeter. Images of printed lattice structures of 20%-30% (w/v) ABS under 40% RH (0.10 mm nozzle at 5 mm/s) are shown in Figure R4(c). It shows that the 30% (w/v) ABS ink has the best shape fidelity, while 20% (w/v) ABS filaments are fused together due to the low viscosity and serious ink spreading. The measured Pr values present in Figure R4(d), which are consistent with the print images. The Pr value is also affected by the RH conditions as seen from R4(e) and (f). For the 25% (w/v) ABS ink being printed at 3 mm/s using a 0.10 mm nozzle, the 70% RH condition results in the best printability factor around 1. The ink cannot have a good printability factor under 40% RH even when the printing speed is reduced from 5 mm/s to 3 mm/s in order to let the ink have sufficient time to solidify and retain a good shape fidelity. Even though printing under RH values of 99% may theoretically yield Pr values close to 1, it must be noted that such conditions are not advisable as the ink may solidify too fast and clog and block the dispensing nozzle, resulting in deformed filaments.

Figure R4. Quantitative evaluation of printing fidelity. (a) Lattice design for printing evaluation. (b) Schematic of printability factor (P_r) measurement. (c) Printing results of 20-30% (w/v) ABS inks under 40% RH using a 100 μm nozzle at 5 mm/s. (d) P_r results for (c). (e) Printing results of 25% ABS ink under 40%, 70%, and 99% RH using a 0.10 mm nozzle at 3 mm/s. (f) P_r results for (e) (scale bars: 5 mm, and N/A: not applicable).

In terms of the print shape accuracy, resulting polymer parts may experience some post-printing shrinkage, given that solvent is extracted out of the deposited filament as a result of the phase separation process. The level of shrinkage depends on the RH value (Figure R5(a) and (b)), polymer concentration (Figure R5(a) and (c)), and polymer material. To study the effect of RH on the shape accuracy, a circular structure with an inner diameter (ID) of 10.0 mm, an out diameter (OD) of 10.9 mm, and a wall thickness of 0.45 mm (Figure R5(a)) is designed and printed to evaluate the print shape accuracy using a gauge 22 nozzle (diameter of 0.41 mm) at 4 mm/s. The 40% (w/v) ABS ink was printed under the 40%, 70%, and 99% RH levels, and the printing results are shown in Figure R5(b). The printed circles are smooth with good shape fidelity, and the measured ID, OD, and wall thickness are close to the design values (R5(d)). Of them, the 40% RH level results in the best print shape accuracy with slight shrinkage.

To study the effect of polymer concentration on the shape accuracy, a circular structure was similarly printed while using the 20%, 30%, and 40% (w/v) ABS inks using a gauge 22 nozzle (diameter of 0.41 mm) at 4 mm/s under 40% RH (R5(c)). The 20% (w/v) ABS ink has severe

spreading due to the low polymeric concentration, resulting in a higher OD and a lower ID than design values, while the 30% and 40% (w/v) ABS inks result in relatively accurate prints due to the combined effect of ink viscosity (for spreading) and solidification speed (R5(e)). Of them, the 40% (w/v) ABS ink results in the best print shape accuracy with slight deviation.

As observed, the good combination of RH and polymer concentration is 70% RH and 30-40% (w/v) ABS during VIPS-3DP ABS printing.

Figure R5. Post-printing solidification-induced filament diameter variation as a function of RH and polymer concentration. (a) Shape design, (b) printing results of 40% (w/v) ABS under 40%, 70%, and 99% RH levels, (c) printing results of 20%, 30%, and 40% (w/v) ABS under 40% RH, and (d-e) measurements of the inner diameter (ID), out diameter (OD), and wall thickness of (b-c), with comparison to the design values (dashed lines).

To study the effect of polymer material on the shape accuracy, both ABS and polyacrylonitrile (PAN) were used as two example polymer materials with different chemical affinities with the solvent (DMSO). Because PAN has a better affinity due to a lower relative energy difference (RED) value than ABS (Supplementary Information Table 1), it leads to a slower phase-separation process. The slow solidification rate of PAN allows it to shrink as it gets slowly solidified and thus, presenting a high post-printing shrinkage degree and smaller pore sizes. In contrast, ABS solidifies fast due to the rapid solvent extraction, having a low post-printing shrinkage degree and forming

highly porous morphology with large pore size. An example of the as-printed and as-solidified parts, 20% (w/v) PAN-based semicircles before and after solidification are shown in Figure R6(a) and (b), presenting a clear reduction in diameter as indicated by black arrows. For comparison in terms of the print shape accuracy, the shrinkage magnitude of the 20% (w/v) ABS and 20% (w/v) PAN inks is quantified to be approximately 3% and 30%, respectively, as seen in Figure R5(b), showing that polymers having a low RED value may shrink more significantly.

Figure R6. (a) Tube printing using 20% (w/v) PAN (scale bar = 10 mm) and (b) shrinkage measurements for ABS and PAN.

By leveraging the printing knowledge mentioned above, we have printed additional structures with improved quality (Figure R7).

Figure R7. VIPS printing demonstrations showing some printed structures with different features and filament diameters. Top left spiral printed using 35% (w/v) ABS with a gauge 32 (diameter of 0.10 mm) under 40% RH. Top middle left bicycle printed using 35% (w/v) ABS with a gauge 32 under 70% RH. Top middle right and top right vases both printed using 35% (w/v) ABS with a gauge 32 under 40% RH. Middle left frog printed using 35% (w/v) ABS with a gauge 27 (diameter of 0.20 mm) under 40% RH. Middle 2D lattice printed using 35% (w/v) ABS with a gauge 27 under 99% RH. Middle right vase printed using 35% (w/v) ABS with a gauge 32 under 70% RH. Bottom 3D lattice printed using 35% (w/v) ABS with a gauge 32 under 99% RH. Insets: structure design. Scale bars: 5 mm (if not specified).

The above information is included as S4 in the revised SI file.

1b) Effect of relative humidity on the part microstructure

Effects of ABS concentrations under different RH levels on the microstructures have also been investigated, and the results are shown in Figure R8. The polymer ABS is chosen for investigations for its proneness to printing-induced porosity when compared to PAN. All the samples show a similar pore distribution, where a dense skin layer is observed on the surface, micropores in the intermediate region, and macropores in the innermost region (SI Figure S5.3). The polymer concentration plays a role in the final porous microstructure formation, and this relationship is well-established [Mulder1996] [Chang2019]. For higher-concentration polymer inks, less amount of solvent needs to be extracted, and this results in a theoretically decreased pore size. The finger-like porous structures are oriented along the solidification direction with distributed porous layers and lamellar dense interlayers. No significant morphological difference is found among different RH levels. However, under a higher RH level, the porous and layers are not easily differentiable due to the faster demixing kinetics.

Figure R8. Morphology of porous structures of ABS with different concentrations and RH levels in the printing environment (Arrows: solidification direction along the layer thickness. Scale bars: 80 μm).

1c) Effect of relative humidity on the shape accuracy and part strength

The strength of the green bodies has been studied during printing the 40% ABS ink under different RH levels (40%, 70%, and 99%) using a gauge 22 (diameter of 0.41 mm) nozzle at 3 mm/s. A dog bone model with a gauge cross section of approximately 1.35 mm (3 filaments in width) and 1.00 mm (4 layers in height) is used for the tensile test [Jin2016] on an eXpert 4000 MicroTester testing system (Admet, Norwood, MA). A stretching speed of 0.05 mm/sec was applied while recording the load on a 1000 g load cell. The low RH allows for the extruded filament to flow and remain liquid for many seconds, resulting in a much higher degree of compositional homogeneity within the printed part and therefore a higher ultimate tensile strength (Figure R9). During this liquid stage prior to solidification, the filaments fuse into one instead of keeping the structure of some connected filament paths. As observed, the 40% RH sample also fractures with a consistent and flat fracture surface perpendicular to the axial strain due to its good ductility. In comparison, the samples fabricated under the 70% and 99% RH are more brittle due to larger internal pores (higher amount of stress concentration regions) and other fabrication-induced defects such as the premature-solidification-induced deformed or broken patterns. The interlayer adhesion under high RH levels is also reduced compared to that under the 40% RH, resulting in a reduced mechanical strength. In particular, premature solidification of the extruded filament under the 99% RH introduces more printing defects, further reducing the mechanical strength and creating a greater

degree of strength variability. It is note that the ABS green body strength is much lower than that of typical ABS products due to the porous microstructure-induced weakening effect. As such, polymeric-based VIPS-3DP printed green bodies are not good for load-bearing applications.

Figure R9. Mechanical strength of 40% (w/v) ABS under 40%, 70%, and 99% RH.

The manuscript has been updated accordingly as follows:

on Page 5:

“One of the key advantages of using VIPS as a solidification mechanism is that such solidification dynamics can be adjusted by controlling the delivery of non-solvent to the printing area, **specifically, the RH level herein**. Printing under room conditions results in overall slower phase-separation dynamics when compared to that under a nebulized printing zone, as the amount of water in the environment is significantly different (RH: 40% vs 99%). **For a given polymeric ink, the printing behavior is highly related to the increased RH from ink spreading to well-defined filament to deformed filament as exemplified in Fig. 2(b1). At low RH levels, the ink is not solidified quickly enough, and thus, there is enough time for the ink to deform due to its own weight and the lack of sufficient surface tension and hardening degree. This effect is most apparent for the low viscosity ink compositions such as 20% w/v ABS since these samples have low shear moduli resulting in poor stability. On the other hand, higher levels of RH may induce premature solidification-induced defects such as wavy filament shapes due to the fast solidification dynamics in place.** Results show that 20%-60% ABS inks under 40% and 70% RH have a larger filament diameter than that of the nozzle (herein a gauge 22 nozzle with an inner diameter of 0.41 mm) as the insufficient solidification after extrusion results in poor shape fidelity, as seen in Fig. 2(b2). **More detailed printability and print shape accuracy studies can be found in SI S4. As observed, the good combination of RH and polymer concentration is 70% RH and 30-40% (w/v) ABS during VIPS-3DP ABS printing. For the demonstration of high-resolution printing using VIPS-3DP, a miniature ABS lattice is printed as shown in Fig. 2(c1), and each segment is approximately 0.30 mm tall (three layers for 0.10 mm each) and 0.30 mm wide (two filaments).”**

and Page 15:

“... Technically speaking, VIPS-3DP provides an alternative DIW approach to printing engineering polymers without heating or tailoring them for certain solidification mechanisms as well as heterogenous composite structures by using polymer as a binder material. The resulting microstructure is porous due to the nature of phase-separation process, which may be utilized for some porosity-inspired applications. For such applications, it should be noted that the pore formation process within the polymeric phase is to be further investigated regarding how the pore size is controlled or affected by the printing process when sacrificial template additives are added during composite printing. In addition, the VIPS-3DP process can be economically implemented by adding a nebulized environment only. Typical DIW processes require the dispensed ink to be solidified *in situ* to retain the printed form. For VIPS-3DP, the solidification rate is bounded by the diffusion speed of non-solvent vapor, polymer demixing rate during phase separation, non-solvent concentration (RH in this study since water is the non-solvent), and polymer concentration. For a given combination of polymer-solvent-non-solvent, the printing speed is limited by the solidification rate of the dispensed ink, and the printing resolution is limited by the size of the smallest nozzle usable for a given polymeric ink as most dissolved engineering polymers are only slightly shear thinning. Furthermore, since the phase separation process is the mechanism for solidification, the resulting polymer parts may experience post-printing variations, for which should be compensated during the design phase.”

In addition, Figure 2 has been updated as follows:

Figure 2. ABS-based printing results: **(a)** sequence of a vase-like structure being printed, **(b1)** schematics of different printed filament morphologies, **(b2)** effects of ABS polymer ink concentrations (20%-60%) on the printed filament diameters under 40%, 70%, and 99% RH (Inset: images of the solidified filaments, and dashed lines in the inset: nozzle diameter. Scale bars: 2 mm), **(c1)** a VIPS-3DP printed lattice part, **(c2)**-(**c3**) morphology details of polymeric cross sections, and **(d)** overhang filament printing process. (Scale bars in (a) and (d): 10 mm).

2. The authors are referred to the following work for practical ideas regarding the tuning of the nebulized environment:

Estelle, K. T., & Gozen, B. A. (2021). Humidity-controlled direct ink writing for micro-additive manufacturing with water-based inks. *Journal of Manufacturing Processes*, 69, 583-592.

Change (s): Thanks to this recommendation, a tunable nebulized environment has been designed and utilized by referring to the setup as recommended [Estelle 2021]. Figure R12(a) shows the overall configuration that consists of the camera, humidifier, humidity controller, and print shroud. The camera has a glass slide placed on the lens to observe the filament printing from the bottom up as seen in Figure R12(b). The print shroud is used to help direct the humid air to the freshly printed ink as well as ensure mixing of the humid and less humid air, which is constructed from a custom designed 3D printed ABS part and a modified 50 mL conical tube as seen in Figure R12(c). The humidity level is tuned by controlling the angle between the outlet of the humidifier and the hose that runs to the print shroud. This angle, θ , is set using the humidity controller as seen in Figure R12(d), and the relative humidity is measured using a HOBO Temp/RH Data Logger (Onset, Bourne, MA). The relationship between the RH levels and the angle θ were experimentally defined.

Figure R12. (A) Overview of the humidity controller setup including the humidifier, humidity controller, print shroud, and the camera. (B) A close-up of the camera with a glass slide printing substrate in place. (C) A close-up of the print shroud constructed from a 3D printed part and a modified 50 mL conical tube. (D) A close-up of the humidity controller. By changing the relative angle, θ , the relative humidity at the nozzle tip can be tuned.

Changes/Responses per the comments from Referee 3

We highly appreciate the encouragement and comments on our study and have updated our manuscript as suggested. In particular, all major changes are highlighted in red ink in the revised manuscript.

1. The authors mentioned that the phase separation speed depends on the properties of the ink. Specifically, the printing speed for the ABS-based and PAN-based ink is 2-5 mm/s and 0.5-2 mm/s, respectively. This printing speed appears to be significantly slower (approximately 10 times) than that of FDM printers. To provide a more comprehensive understanding of the scalability of the VIPS technique, it would be beneficial if the authors could discuss the potential challenges associated with the slower printing speed. Additionally, could the authors suggest any strategies or future research directions for improving the speed while maintaining the quality of prints?

Change(s): As pointed out by the reviewer, the printing/path speed is relatively low when comparing with that of other printing processes, which is due to the VIPS-based solidification physics. Once ink is dispensed out of the nozzle, it needs to be solidified *in situ* to retain its shape within a given timescale to ensure proper bonding is ensured. For such a solidification dynamics analysis, the solidification timescale is estimated via optical measurements, and Figure R1 presents an example, where it is shown a typical 20% (w/v) ABS-based ink being printed on a glass substrate and observed (bottom view) over time to quantify the solidification rate during the VIPS printing process by capturing the opacification of the filament. For the 20% (w/v) ink under 40% RH, it takes approximately 90 seconds for the opacification to be fully completed. The printing speed needs to match the solidification rate for a good shape fidelity and proper inter-layer fusion. For the 20% ink under higher RH, the opacification is a much more rapid process (<10 seconds as this is the time it takes for the contrast to adjust with our imaging setup). It should be pointed out that higher concentration ABS inks present significantly more opacity in the liquid phase, inhibiting any meaningful observation of the change in opacity over time. The information is included as S3 in the revised SI file, and the manuscript has been updated on Page 5 as follows:

“... This leads to slower demixing kinetics of PAN upon exposure to the non-solvent (water) as seen from the different polymer-solvent interaction parameters g_{23} as discussed in SI S2.2. **Typical evolution of the solidification front after filament printing can be seen in SI S3. ...**”

Figure R1. Evolution of the solidification front of 20% ABS ink at 40% RH over time showing 0, 30, 60, 75, and 90 seconds after filament printing. Scale bar is 2 mm.

In general, this solidification rate is bounded by the diffusion speed of non-solvent vapor, polymer demixing rate during phase separation, non-solvent concentration (relative humidity (RH) in this study since water is the non-solvent), and polymer concentration. For a given combination of polymer-solvent-non-solvent, the printing speed cannot be increased without bound in order for the dispensed ink to solidify. Such a slower printing speed may result in low productivity. For possible high printing speeds, it may be beneficial to try the following strategies:

- 1) Use of polymers with a high concentration and/or high demixing rate: To investigate and develop ink formulations that facilitate faster demixing and phase separation without compromising the print quality, and such a goal could be achieved via increasing polymer concentration or selecting polymers with a high relative energy difference (RED) value with a given solvent.
- 2) Implementation of parallel printing setup: To focus on advancements in extrusion systems and nozzle design that could reduce the deposited volume per dispensing head such as the use of multi-nozzle or multi-printhead configurations to enable parallel printing. This may allow multiple layers or sections of an object to be printed simultaneously to increase the printing productivity.

The manuscript has been updated accordingly on Page 15 as follows:

“... For VIPS-3DP, the solidification rate is bounded by the diffusion speed of non-solvent vapor, polymer demixing rate during phase separation, non-solvent concentration (RH in this study since water is the non-solvent), and polymer concentration. For a given combination of polymer-solvent-non-solvent, the printing speed is limited by the solidification rate of the dispensed ink, ...”

2. While the paper presents compelling case for the versatility of the VIPS technique across various materials, it is important to address the advantages of using VIPS for metal printing in comparison to established commercial metal printing technique such as binder jetting. Please provide a more

detailed discussion highlighting the unique strengths of VIPS for metal printing, especially regarding factors like cost-effectiveness, material compatibility or potential applications where VIPS outperforms existing methods.

Change(s): As noted by the reviewer, metal additive manufacturing (AM) can be implemented in different ways, and each technology has its own specified applications and limitations. Based on the printing temperature (whether at elevated temperatures), metal AM can be classified as: high-temperature printing and binder-assisted low-temperature printing. The former mainly includes powder bed fusion and directed energy deposition. The latter is generally binder assisted but requires a post-printing sintering step to sinter metal powders, including the proposed VIPS-3DP.

The binder-assisted printing processes can be further classified based on whether a powder bed is utilized during the printing of green parts: 1) powder-bed based: binder jetting, and 2) without a powder bed such as the bound powder extrusion (BPE)/metal material extrusion. Different from the aforementioned binder-assisted metal printing technologies without a powder bed, VIPS-3DP is proposed based on the physics that polymeric structures can be effectively fabricated by various phase inversion (or phase separation) methods based on the thermodynamic instability provoked from a homogeneous polymeric system by temperature and/or composition change. The build material for VIPS-3DP metal printing is a metal-polymer suspension ink made of metallic powders and a dissolved polymer, with the latter functioning as a binder material, which is removed via polymer thermal debinding after metal-polymer green parts are printed. Specifically, VIPS-3DP enables the freeform fabrication of metal-polymer structures from metal-polymer suspensions spatially at any metal composition at room temperature by using a mixer. Once a metal-polymer green part is obtained, it further undergoes polymer thermal debinding and metal powder sintering cycles to burn out the binder-functioning polymer and induce solid-state sintering to the metal powders, sequentially, resulting in a fully sintered metallic part. As needed, the resultant metallic part can be further infiltrated with a second-phase metal to mitigate the porosity (the percentage of total void space).

The novelty of the VIPS-3D metal printing technology is further summarized as follows:

- A convenient, low-cost printing setup by only adding a nebulized environment to a generic direct ink writing (DIW) printer to obtain green bodies,
- A unique solidification mechanism (vapor-induced phase separation) is utilized to solidify metal-polymer features being printed at room temperature,
- It is capable of printing heterogeneous metallic or metallic composite structures by using a pre-printing ink mixer, which mixes different metallic powder and other phase inks *in situ* at any ratio for printing, and
- A unique solvent-enabled printing process where the used solvent can be reclaimed and reused to reduce the environmental impact of the printing process.

The main manuscript has been updated on Page 7 as follows:

“...solidus point of the alloy (SI S6.1). **One of the key advantages of VIPS-3DP when compared to binder-based metal printing technologies is the convenient, low-cost printing setup at room temperature and the ability to efficiently tune the composition**

of the deposition by easily changing the input composition (e.g. integrating with an active pre-printing ink mixer)."

3. In the VIPS process, the outer region is solidified first. The authors claim that "such a solidification process is intended to present dissolved polymer to ensure adjacent interlayer fusion". Since the fusion happens at the outer region of the filament, which is solidified, why it can ensure the bonding?

Change(s): As noted by the reviewer, the outer region is solidified first during VIPS-3DP. From a single filament perspective, it is true that the outermost layer solidifies first if exposed to a non-solvent environment. Fortunately, part of the outer region interacts with a consecutively deposited filament that contains solvent, and this solvent may dissolve the outer region (of the previous filament) being contacted and further form a good fusion bond.

The manuscript has been updated on Page 4 as follows:

"From a 3D printing perspective, such a solidification process is intended to occur only partially to guarantee enough stiffness to allow the structural buildup while presenting enough dissolved polymer to ensure adjacent interlayer fusion. **The solidified outermost layer of a previous filament is partially dissolved by the solvent of a consecutively deposited filament, forming a good fusion bond. ..."**

During the discussion on the morphology and porosity of polymeric filaments in the SI, we have also stated as follows (S3):

"Faster phase-separation kinetics promote the formation of hardening skin layer that leads to consecutively deposited layers not being fused well due to the significantly different hardening degree of their contact surfaces. If a previously deposited layer has a notable solvent-extraction time in the outer surface layer under fast phase-separation kinetics, it may harden well and form a dense skin layer before a subsequent filament is deposited. The difference in hardening status and the presence of a thin layer of solvent and non-solvent may result in insufficient bonding between two consecutive layers and a visible interface."

4. The authors claim that the solvent can be collected. Please provide detailed information on how the solvent in the nebulized non-solvent vapor can be collected. Additionally, it would be valuable to discuss any potential environmental concerns regarding the release of dissolved solvents into the printing environment.

Change(s): Different from other evaporation-based phase-separation processes, there should be no vapor formed during the proposed VIPS-3DP process since we use a solvent with a low vapor pressure but good affinity to non-solvent (water in this study). If needed, a printing chamber can be used.

In general, VIPS requires a hydrophobic surface by which mixed solvent (DMSO) and water can be easily collected upon printing without compromising the printed structure. The resultant liquid non-solvent and solvent mixture is collected from the print substrate, and the collect solvent (DMSO) and non-solvent (water) mixture is processed to reclaim DMSO based on the distillation

mechanics, which is enabled by the evaporation temperature difference (100°C for water vs. 189°C for DMSO). As perceived, there is no potential environmental concern with VIPS-3DP when using the DMSO-water combination.

The main manuscript has been updated on Page 4 as follows:

“... The resulting solvent can be collected and further reclaimed accordingly for environmentally conscious manufacturing. **It is also expected that solvents with a low vapor pressure should be utilized to avoid potential evaporation during printing and applicable hydrophobic or hydrophilic substrate surface should be used for better collection of the resultant solvent and non-solvent mixture upon printing without compromising printed structures. ...**”

References:

- [Chang2019] Chang, H.-Y. & Venault, A. Adjusting the morphology of poly(vinylidene fluoride-co hexafluoropropylene) membranes by the vips process for efficient oil-rich emulsion separation. *Journal of Membrane Science* **581**, 178–194 (2019).
- [Estelle2021] Estelle, K. T. & Arda Gozen, B. Humidity-controlled direct ink writing for micro-additive manufacturing with water-based inks. *Journal of Manufacturing Processes* **69**, 583–592 (2021).
- [Jin2016] Jin, Y., Compaan, A., Bhattacharjee, T. & Huang, Y. Granular gel support-enabled extrusion of three-dimensional alginate and cellular structures. *Biofabrication* **8**, 025016 (2016).
- [Mulder1996] Mulder, M. *Basic principles of membrane technology*. (Kluwer Academic Publishers, 1996)
- [Nino-Amezquita2010] Nino-Amezquita, O. G., Enders, S., Jaeger, P. T. & Eggers, R. Measurement and prediction of interfacial tension of binary mixtures. *Industrial & Engineering Chemistry Research* **49**, 592–601 (2009).
- [Ren2023] Ren, B., Song, K., Chen, Y., Murfee, W. L. & Huang, Y. Laponite nanoclay-modified sacrificial composite ink for perfusable channel creation via embedded 3D printing. *Composites Part B: Engineering* **263**, 110851 (2023).
- [Soltan2019] Soltan, N., Ning, L., Mohabatpour, F., Papagerakis, P. & Chen, X. Printability and cell viability in bioprinting alginate dialdehyde-gelatin scaffolds. *ACS Biomaterials Science & Engineering* **5**, 2976–2987 (2019).
- [Zhang2020] Zhang, P., Wang, Z., Li, J., Li, X. & Cheng, L. From materials to devices using fused deposition modeling: A state-of-art review. *Nanotechnology Reviews* **9**, 1594–1609 (2020).

REVIEWER COMMENTS

Reviewer #1 (Remarks to the Author):

see attachment

Reviewer #2 (Remarks to the Author):

The authors addressed my primary concerns. I recommend the manuscript for publication.

Reviewer #3 (Remarks to the Author):

The authors have thoughtfully responded to the reviewers' concerns. I am satisfied with the revision.

Review for Nature Communications

Title: Vapor-induced phase separation enabled versatile direct ink writing

Authors: Sole-Gras et al.

This review refers to the revised version of the manuscript I received on Dec 12, 2023. Generally, the authors have addressed my questions and concerns in a careful and thorough manner. However, several issues listed below remain to be clarified and corrected prior to publication of the manuscript:

- 1) The authors mention porosity and pore size of the printed struts referring to scanning electron microscopy images. This is an important feature relevant for the mechanical strength of the printed objects. Therefore, the authors must provide quantitative information about porosity and pore size (as a function of polymer concentration and RM).
- 2) On page 4 of the response letter and in the revised manuscript the authors claim “most dissolved engineering polymers are only slightly shear thinning”. This general statement is definitely wrong and can be deleted without loss.
- 3) In the same chapter the authors discuss the effect of demixing rate and other printing parameters on the printing results. Again, this is done in a qualitative manner but should be supported by exemplary numbers.
- 4) The results shown in Fig R3b are not consistent with the corresponding text. The authors say that a nozzle with an inner diameter of 0.41 mm has been used for printing. The images inserted in R3b show that e.g. for RH = 99 % the diameter of the printed filaments only slightly exceeds that of the nozzle. The diameter values shown in the graph are, however, close to 2 mm. This apparent contradiction must be corrected.
- 5) Figs. R5 d,e) would be easier to comprehend if the experimental data were normalized to the respective design values.
- 6) On page 8 of the response letter the authors again refer to “highly porous morphology with large pore sizes” without providing numbers or any other quantitative information.
- 7) On page 11, line 4 and 5 the authors admit that “polymeric-based VIPS-3DP printed green bodies are not good for load-bearing applications”. This important feature must be explicitly mentioned in the manuscript.
- 8) On page 14 of the response letter and in the revised manuscript the authors claim “Most ceramic or metallic particle-based suspensions are not yield-stress fluids” This general statement again is wrong and must be deleted. The example shown in Fig. R11 is not helpful. Of course, the flow properties also of ceramic or metallic particle suspensions strongly depend on particle volume fraction, particle size (distribution) and so-called colloidal or thermodynamic interactions among particles. None of these parameters is specified for suspensions to which Fig. R11 refers. Data for the corresponding pure polymer solution are also missing.

List of Changes / Responses Letter

Article ID: NCOMMS-23-03487A

Title: Vapor-Induced Phase Separation-Enabled Versatile Direct Ink Writing

Changes/Responses per the comments from Referee 1

We highly appreciate the encouragement and comments on our study and have updated our manuscript as suggested. In particular, all major changes are highlighted in red ink in the revised manuscript.

1. The authors mention porosity and pore size of the printed struts referring to scanning electron microscopy images. This is an important feature relevant for the mechanical strength of the printed objects. Therefore, the authors must provide quantitative information about porosity and pore size (as a function of polymer concentration and RH).

Change(s): As suggested, we have further investigated the overall porosity and pore size as functions of ABS concentration and relative humidity (RH) level, respectively. Since the VIPS process results in a typical phase separation-related microstructure (i.e. a dense skin layer on the surface, micro-voids/micro-pores in the intermediate region, and macro-voids/macro-pores in the innermost region (SI Figure S9)), we have quantified the overall porosity and pore size of entire filaments. The analysis has been conducted using ImageJ software based on the SEM images.

As seen in Figure R1 (Figure S10 in the SI file), for the selected ABS concentrations (40% and 60%, typical concentrations used for polymer printing in this research) and RH conditions (40%, 70%, and 99%) investigated, the overall porosity varies between 35% and 45%, and there is no significant difference observed among these samples (Figure R1 (a)). The majority of the observed voids on the cross section of whole filaments has a pore size between 1 and 5 μm (Figure R1 (b)). Although no change in the overall porosity is observed as the RH level increases, the average pore size decreases as observed in the distribution plots (Figure R1 (b1) (from 2.78 to 2.59 to 2.34 μm) and (b2) (from 2.81 to 2.50 to 1.87 μm)) wherein the smaller voids tend to have a higher frequency. This phenomenon is observed for both 40% and 60% (w/v) ABS, and the decrease in the pore size is more pronounced with the 60% (w/v) ABS, meaning that higher concentration polymers such as ABS may be more prone to form smaller voids under higher RH levels.

Figure R1 (Figure S10 in the SI file). (a) Porosity of 40% and 60% (w/v) ABS, and (b) pore size distribution of (b1) 40% and (b2) 60% (w/v) ABS under different RH levels (40%, 70%, and 99%) using a 0.41 mm nozzle at 1.0 mm/s. (“NS” indicates not significant ($p > 0.05$) in the porosity value between the samples (paired t test).)

These results have been included into “S5 Morphology and porosity of polymeric filaments” in the revised SI file as follows:

On Pages 11,12, and 14 in the SI file, it has been updated as follows:

“As seen in Fig. S10, for the selected ABS concentrations (40% and 60%, typical concentrations used for polymer printing in this research) and RH conditions (40%, 70%, and 99%) investigated, the overall porosity varies between 35% and 45%, and there is no significant difference observed among these samples (Fig. S10 (a)). Most of voids found on the cross section of whole filaments present a pore size between 1 and 5 μm (Fig. S10 (b)). Although no change in the overall porosity is observed as the RH level increases, the average pore size decreases as observed in the distribution plots (Fig. S10 (b1) (from 2.78 to 2.59 to 2.34 μm) and (b2) (from 2.81 to 2.50 to 1.87 μm)) wherein the smaller voids tend to have a higher frequency. This phenomenon is observed for both 40% and 60% (w/v) ABS, and the decrease in the pore size is more pronounced with the 60% (w/v) ABS, meaning that higher concentration polymers such as ABS may be more prone to form smaller voids under higher RH levels.

Figure S10. (a) Porosity of 40% and 60% (w/v) ABS, and (b) pore size distribution of (b1) 40% and (b2) 60% (w/v) ABS under different RH levels (40%, 70%, and 99%) using a 0.41 mm nozzle at 1.0 mm/s. (“NS” indicates not significant ($p > 0.05$) in the porosity value between the samples (paired t test).)”

2. On page 4 of the response letter and in the revised manuscript the authors claim “most dissolved engineering polymers are only slightly shear thinning”. This general statement is definitely wrong and can be deleted without loss.

Change(s): As suggested, this statement has been deleted from the manuscript.

3. In the same chapter the authors discuss the effect of demixing rate and other printing parameters on the printing results. Again, this is done in a qualitative manner but should be supported by exemplary numbers.

Change(s): As suggested, the effects of RH levels on the ink demixing rate and printing accuracy have been quantitatively evaluated, and the results are shown in Figure R2 (Figures S3 in the updated SI file). During the VIPS process, the demixing rate is dependent on the polymer concentration and RH level, where a higher polymer concentration and a higher RH level lead to a higher demixing rate (Figure R2 (a)). The solidification rate k ($k = \frac{\text{Solidification front thickness } (\mu\text{m})}{\text{Time (s)}}$) increases from 0.02 $\mu\text{m/s}$ to 0.26 $\mu\text{m/s}$ for the 20% (w/v) ABS as the RH level increases from 40% to 99% (Figure R2 (a-b)). A higher demixing rate allows the deposited ink to solidify faster and hold the

shape promptly, while a lower demixing rate allows the ink to spread, leading to a flat shape with a larger diameter. As shown in Figure R2 (b), the spreading of the initially deposited filament leads to a larger diameter of a solidified filament at low RH levels (such as 40% and 70%), which is due to the slower demixing rate. At 99% RH, the diameter variation between the nozzle diameter and solidified filament significantly decreases, indicating less ink spreading and a better printing resolution.

It is noted that only 20% (w/v) ABS was investigated for its reasonable opaqueness level that resulted in optimized contrasts for the imaging of the evolution of solidification front.

Figure R2 (Figure S3 in the SI file). (a) Evaluation of the solidification front of 20% (w/v) ABS ink under 40%, 70%, and 99% RH over time using a gauge 22 nozzle (diameter of 0.41 mm). (b) Effect of RH (40%-99%) on the printed 20% (w/v) ABS filament diameter using a gauge 15 nozzle (diameter of 1.36 mm) and a printing speed of 1.5 mm/s and effect of RH on the solidification rate k .

The title of S3 of the SI file has been revised as follows: “S3 Evolution of the solidification front **and effect of demixing rate on the filament diameter**”.

In addition, Pages 5 and 6 of S3 has been updated accordingly as follows:

“During the VIPS process, the demixing rate is dependent on the polymer concentration and RH level, where a higher polymer concentration and a higher RH level lead to a higher demixing rate (Figure S3 (a)). The solidification rate k ($k = \frac{\text{Solidification front thickness } (\mu\text{m})}{\text{Time } (s)}$) increases from 0.02 $\mu\text{m/s}$ to 0.26 $\mu\text{m/s}$ for the 20% (w/v) ABS as the RH level increases from 40% to 99% (Figure S3 (a-b)). A higher demixing rate allows the deposited ink to solidify faster and hold the shape promptly, while a lower demixing rate allows the ink to spread, leading to a flat shape with a larger diameter. As shown in Figure S3 (b), the spreading of the initially deposited filament leads to a larger diameter of a solidified filament at low RH levels (such as 40% and 70%), which is due to the slower demixing rate. At 99% RH, the diameter variation between the nozzle diameter and solidified filament significantly decreases, indicating less ink spreading and a better printing resolution.

Figure S3. (a) Evaluation of the solidification front of 20% (w/v) ABS ink under 40%, 70%, and 99% RH over time using a gauge 22 nozzle (diameter of 0.41 mm). (b) Effect of RH (40%-99%) on the printed 20% (w/v) ABS filament diameter using a gauge 15 nozzle (diameter of 1.36 mm) and a printing speed of 1.5 mm/s and effect of RH on the solidification rate k .”

4. The results shown in Fig R3b are not consistent with the corresponding text. The authors say that a nozzle with an inner diameter of 0.41 mm has been used for printing. The images inserted in R3b show that e.g. for RH = 99 % the diameter of the printed filaments only slightly exceeds that of the nozzle. The diameter values shown in the graph are, however, close to 2 mm. This apparent contradiction must be corrected.

Change(s): Thanks for pointing the typo out, this has been corrected in the manuscript as follows as the wrong nozzle diameter was referenced:

On Page 5:

“Results show that 20%-60% (w/v) ABS inks under 40% and 70% RH have a larger filament diameter than that of the nozzle (**herein a gauge 15 nozzle with an inner diameter of 1.36 mm**) as the insufficient solidification after extrusion results in poor shape fidelity, as seen in Fig. 2(b2).”

5. Figs. R5 d,e) would be easier to comprehend if the experimental data were normalized to the respective design values.

Change(s): As suggested by the reviewer, Figure S5 in the SI file has been replaced as follows by including the normalized values of the internal (ID) and external diameters (OD) of the printed cylinders:

Figure R4 (Figure S5 in the SI file). Post-printing solidification-induced filament diameter variation as a function of RH and polymer concentration. (a) Shape design, (b) printing results of 40% ABS under 40%, 70%, and 99% RH levels, (c) printing results of 20%, 30%, and 40% ABS under 40% RH, and (d-e) measurements of the inner diameter (ID) and outer diameter (OD) of (b-c), **normalized to the design values**.

6. On page 8 of the response letter the authors again refer to “highly porous morphology with large pore sizes” without providing numbers or any other quantitative information.

Change(s): This general statement has been deleted from the sentence since it has been concluded indirectly based on the final filament diameter. The detailed discussion on the pore size and porosity has been included in the revised SI file following the reviewer’s Comment #1.

7. On page 11, line 4 and 5 the authors admit that “polymeric-based VIPS-3DP printed green bodies are not good for load-bearing applications”. This important feature must be explicitly mentioned in the manuscript.

Change(s): As suggested by the reviewer, this statement has been included into the manuscript as follow:

On Page 15:

“The resulting microstructure is porous due to the nature of phase-separation process, which may be utilized for some porosity-inspired applications **but not intended for load-bearing applications directly.**”

8. On page 14 of the response letter and in the revised manuscript the authors claim “Most ceramic or metallic particle-based suspensions are not yield-stress fluids” This general statement again is wrong and must be deleted. The example shown in Fig. R11 is not helpful. Of course, the flow properties also of ceramic or metallic particle suspensions strongly depend on particle volume fraction, particle size (distribution) and so-called colloidal or thermodynamic interactions among particles. None of these parameters is specified for suspensions to which Fig. R11 refers. Data for the corresponding pure polymer solution are also missing.

Change(s): As suggested by the reviewer, the mentioned statement has been removed from the manuscript.

The detailed information on the metallic ink composition and particle information are provided on Pages 7 and 16 of the manuscript as follows:

On Page 7:

“Powders herein are suspended in the dissolved polymer with a volumetric ratio within the range of 50-55%, close to the random close-packing (RCP) ratio of hard, equally sized particles of 64%¹⁴.”

and Page 16:

“For 316L printing, metallic suspensions were prepared by mixing the ABS and PAN polymer solutions, respectively, with stainless-steel 316L powders (D10: 4.61 μm, D50: 12.58 μm, D90: 24.44 μm, MSE Supplies, Tucson, AZ, USA).”

Changes/Responses per the comments from Referee 2

The authors addressed my primary concerns. I recommend the manuscript for publication.

Thank you for all your suggestions for this manuscript.

Changes/Responses per the comments from Referee 3

The authors have thoughtfully responded to the reviewers' concerns. I am satisfied with the revision.

Thank you for all your suggestions for this manuscript.

REVIEWERS' COMMENTS

Reviewer #1 (Remarks to the Author):

The authors have perfectly addressed my questions and concerns listed in the previous review in their revised manuscript, which from my point of view now is ready for publication.